# Just Energy Transition of South Africa in a Post-COVID Era

Heinrich R. Bohlmann [1], Jessika A. Bohlmann [2], Margaret Chitiga-Mabugu [2] and Roula Inglesi-Lotz [1,*]

1 Department of Economics, University of Pretoria, Pretoria 0002, South Africa; heinrich.bohlmann@up.ac.za
2 Faculty of Economic and Management Sciences, Main Campus, University of Pretoria,
  Pretoria 0002, South Africa; jessika.bohlmann@up.ac.za (J.A.B.); margaret.chitiga@up.ac.za (M.C.-M.)
* Correspondence: roula.inglesi-lotz@up.ac.za

**Abstract:** The impacts of the COVID-19 pandemic have sparked global debate over how green economic recovery may and should be, and if the pandemic has accelerated the present energy transition while assuring a just transition for vulnerable populations such as unskilled workers and women. This study investigates the socioeconomic impact of South Africa's planned green energy transition, with a focus on the Mpumalanga province—the country's largest coal mining region with many coal-fired power plants. Using a regional-dynamic computable general equilibrium (CGE) model, the study analyses the economy-wide effects of different policy scenarios related to a changing electricity generation mix, investment financing costs, and international action against non-compliant industries, amongst others, with a specific focus on the vulnerable industries and population groups in Mpumalanga. Key results from the study highlights that (1) the structure of the Mpumalanga economy will be affected in the medium to long run regardless of the domestic transition path, (2) the Mpumalanga economy is indeed in danger of shrinking relative to the baseline, unless the Just Energy Transition (JET) is quickly and carefully managed, and (3) at a national level, at least, there is the strong possibility of a double dividend when greening the South African economy with overall economic growth and environmental outcomes expected to improve in the long run.

**Keywords:** Just Energy Transition; Leave No One Behind; sustainability; greening; coal

## 1. Introduction

The COVID-19 pandemic led to a global economic crisis of extensive magnitudes, which affected economies, societies, and health systems across the globe [1]. Given the unprecedented situation brought by COVID-19, the overall focus of governments and civil society was to get the disease under control and revive their economies [1]. During the lockdown periods, global economic activity was reduced. Consumers and suppliers substantially altered their profiles and trends, as demonstrated by the short-run halting of greenhouse gas (GHG) emissions and reduced energy usage. As waves of the pandemic continued to roll worldwide in 2020 and 2021, different stimulus packages and vaccine rollouts allowed most economic activity to return. The global energy demand reportedly rebounded by over 4.5% in 2021—above pre-pandemic levels.

The COVID-19 epidemic has sparked a major global debate about the level of environmental sustainability that can be achieved throughout the economic recovery phase. This discussion also explores how much the epidemic has accelerated the transition to sustainable energy sources while simultaneously assuring an equitable transition for vulnerable populations, particularly women and unskilled labourers. For South Africa, one of the significant challenges remains to substitute fossil fuel consumption, which represents over 90% of the energy sources of the country, especially in vital economic sectors such as mining, iron, and steel [2]. The dependence of the sector on energy and capital makes them adapt more slowly to change, and these sectors are also recognised as key employers in the country [3]. Another challenge facing middle- and low-income countries is that

universal access to energy during transition conditions is imperative to achieve the developmental goals of the countries (see United Nations Sustainable Development Goal SDG 7). Such a combination, which includes transitioning to cleaner fuels to mitigate climate change while ensuring that vulnerable populations will mostly be positively affected (or at least not negatively affected), is referred to in the literature as the concept of just energy transition [4,5].

The short- and long-run consequences of the COVID-19 pandemic have not affected everyone in a homogenous way [6]. Chitiga et al. [7] show that the impact of the responses to limit the spread of COVID-19 in South Africa, such as lockdowns and restricted international travel, had devastating effects on several sectors of the economy. Gross Domestic Product (GDP) substantially fell, and employment was decimated. It was found in the study that the most harshly affected were unskilled workers, those with primary school education or less. This finding of the heavier impact on unskilled workers is important, because these workers could not telework like other, more skilled workers. Women were also among the most negatively affected as they tended to be employed in service sectors, such as tourism, which suffered extended total closures [7]. The results showed that the ultra-poor, who usually depended on social grants, which continued during COVID-19, were somewhat shielded and could stay at the same poverty levels while a new poor group emerged. One of the recommendations from these studies was that it was important for policymakers to ensure that this new poor group, consisting mainly of unskilled workers and women, should not be left behind in crafting stimulus policies.

The energy transition process aims at reducing greenhouse gas emissions as the main target using as input the investment in technological changes. As such, the indirect impacts are demonstrated as losses or gains in economic welfare or job opportunities. Economic sectors react differently, offer different acceleration options, or contribute to the energy transition. Evaluating the short and long-run impacts of the just energy transition among various economic sectors is essential for policymakers to prioritise those with the highest net result (minimising both financial and socioeconomic losses) and consider the principle of Leaving No One Behind (LNOB). The principle LNOB is based on moving away from assessing progress on averages and means but ensuring that all population groups progress. It thus becomes imperative for society, particularly researchers, civil society, and policymakers, to compare and contrast the disaggregate progress of all population groups in all aspects. People left behind in development are often economically, socially, spatially, and politically excluded—for example, due to ethnicity, race, gender, educational attainment, age, disability, or a combination of these, leading to multiple discriminations.

According to the United Nations Development Programme [8], there are five factors of LNOB to be considered in assessing the evidence of who is left behind and to what degree: (1) Discrimination; (2) Geography; (3) Vulnerability to shocks; (4) Governance; (5) Socioeconomic status. However, these factors are not isolated but are interdependent in actual conditions. Significant structural socioeconomic changes, such as the energy transition, create different conditions for population groups that have established their livelihoods for decades and have managed to stay away from the poverty line. These new economic rules and policies might threaten some of these conditions if not properly understood and planned. Due to their geographic location and proximity to electricity generation or skills, some population groups are at risk of lacking opportunities and capabilities to equitably participate in the economy. Such conditions might also hinder the ability of these population groups to earn an adequate income. Creating or perpetuating inequalities can cause significant problems for economic prosperity. Promoting the energy transition in South Africa in the name of environmental sustainability while creating a growing number of populations excluded from the labour market or trapped in unsatisfactory, low-waged jobs seems like an unfair trade-off of problematic situations. The concept of LNOB does not advocate the halt of progress but, on the contrary, takes a holistic view of all aspects of transition. This study uses the regional computable general equilibrium (CGE)

model. This economy-wide modelling tool allows us to consider the regional, sectoral, and socioeconomic aspects of the LNOB agenda considering the post-COVID era.

Expanding on the views from the research done by Van Heerden et al. [9], South African energy policymakers face a triple challenge: (1) to generate adequate energy for the growing demands of the population; (2) to develop and use clean energies to reduce greenhouse gas (GHG) emissions; and (3) to minimize the socioeconomic losses from the changes (job, income, and trade losses for example). Such a triple-dividend challenge is common to South Africa's case or a developing country's conditions. Developed and highly industrialized economies, such as Germany, encountered difficulties in their path to a cleaner energy mix. The example case of Energiewende is the ongoing transition to a low-carbon, environmentally sound, reliable, and affordable energy supply that relies on renewable energy [10]. Aiming at a 100% renewable energy system, the German energy transition plan had to consider losses from the status quo changes, particularly in the coal mining sector.

Policies must be appropriately directed and concerted to balance the triple challenge elements effectively. Furthermore, the challenge is multifaceted and exhibits at least two layers of depth: geographic and sectorial differentials. This study aims to provide insights into how the continuous energy transition will influence the socioeconomic conditions of the South African population and how policies can be proactive. The study aims to cover all provinces in South Africa regarding geographical differences. Sectorial differences will be examined in the study, considering that economic sectors will enter the energy transition and will also be affected by national policies and international agreements guiding their transition. This study's purpose is to systematically evaluate a green energy transition whilst considering the LNOB principle. This will ensure that policies that will have the most significant long-term economic impact for all, including the most vulnerable population groups of South Africa, are evaluated while addressing the transition towards cleaner energy sources and ensuring sustainable access to energy for all in a post-COVID era.

As evidenced by many countries supportive of climate-related agreements, most notably via the Paris Agreement adopted at COP21 in 2015, there is widespread consensus on the need and the urgency of a global energy transition towards low-carbon technologies. As with any status quo change, the establishment will experience wins and losses. Recent literature examines the quantitative impact of such a transition and, at the same time, shows that the outcomes of the transition will not only be on the carbon emissions (the initially intended impact of the energy transition) but also on the socioeconomic conditions of the population.

This study contributes to the global and national debates on the greening of economies. The discussion extends from whether such a transition is necessary to how it is designed and implemented to maximise environmental benefits while minimising socioeconomic losses. The relevance of such a study is more prominent for the economies during the recovery phase from the pandemic, notably because world leaders once again agreed to strengthen and reinforce climate change policies. This study and its approach make the following contributions from an academic and policy perspective:

- Methodologically, the study uses a regional-dynamic computable general equilibrium (CGE) model, which is an advanced technique well-suited to broadening quantitative macroeconomic insights across multiple variables in an integrated economy-wide framework.

- From a policy perspective, the CGE modelling approach considers the direct and indirect impacts of such a transition in both the short- and long-run. Such information, and a combination thereof, may affect policymakers and lead to changes to current policies or even result in new guidelines. Examining the effects of the proposed energy transition at only a national level would ignore disparate regional outcomes; as such, this study will offer recommendations for future policies at a provincial level, especially considering the significant inequalities (income, poverty, education, infrastructure etc.) among them.

## 2. Literature Review

The energy transition is not a novel concept in the history of humankind. The interlinkage between humans and energy has changed over time, for example, from traditional energy sources to modern ones during the waves of the rapid industrial revolution [11], while the ongoing interlinkage focuses on the substitution of exhaustible "brown" energies by renewable "green" energies [12]. The necessity for the recent energy transition comes at a time when the world is undergoing broader socioeconomic transformations and transitions, as well as changes in market structures towards liberalisation, on top of the realisation that dealing with climate change is imperative [13]. One of the hurdles in the transition to cleaner fuels is the differences between the energy use patterns of various system users. The high dependence on "brown" energies by the most energy-intensive economic sectors, such as mining, iron, steel, and other metals, makes them cumbersome and inflexible in their transition [3].

The concept of "energy transition" depicts a gradual movement away from carbon technologies dominating the energy systems. At the same time, the word "Just" stresses the importance that this transition should be in such a manner that will not negatively affect society and livelihoods. The United Nations Sustainable Development Goal 7 stipulates that the countries aim to provide clean, affordable, and reliable energy for all. However, the world is moving towards this goal at two different speeds. Industrialised countries with well-established energy networks and policies have an advantage in generating and distributing energy for all and promoting energy as a public good. Conversely, emerging economies, especially those of the Global South, struggle to provide reliable and affordable energy to their rising populations, let alone be able to consider their environmental consequences [14]. Renewable energy technologies are suggested as the most flexible solution to increase the energy access of the world. Bomberg and McEwen [15] indicate that including communities in designing and implementing energy projects is often presented as an intervention to tackle energy poverty—and potentially economic poverty. Such an inclusion can also provide an opportunity to improve energy justice, as per McCauley et al. [16].

Although the transition has documented socioeconomic and environmental benefits, it will not be without physical implications, such as additional generation and investment [17]. In recent years, the attention of the analysis on the socioeconomic impacts of the energy transition is focused on the employment impacts of the shift. As stated by Ram et al. [17], transitioning to a low-carbon economy will create, replace, eliminate, and transform existing jobs like other economic and technological changes. Sheikh et al. [18] noted that the renewable energy sector demonstrates the potential for job creation.

In South Africa, the potential for job creation by the transition is a complex issue with many aspects. The first concerns the structural nature of the jobs as per the Integrated Energy Plan (IEP) by DoE (2016). Table 1 below presents the identified categories. For this analysis, "the measurement used across technologies for both capital expenditure (capex) and operating expenditure (opex) jobs is the number of 'job-years' that the total expenditure per plant creates" [19]. Ram et al. [17] quantify the total direct jobs as "a sum of jobs in manufacturing, construction and installation, operations and maintenance, fuel supply associated with electricity generation, decommissioning of power plants at the end of their lifetimes and transmission".

**Table 1.** Structure of the job-creation-potential assessment.

| | Direct Jobs | Supplier Jobs |
| --- | --- | --- |
| Capex jobs | Direct jobs related to the construction of the power station, for example:<br>• Developers<br>• Planners<br>• Construction workers<br>• Bricklayers | Supplier jobs related to the construction of the power station, for example:<br>• Turbine manufacturers<br>• Solar PV panel manufacturers<br>• Cement producers<br>• Steel producers |
| Opex jobs | Direct jobs related to the operation of the power station, for example:<br>• Power plant workers<br>• Coal mine workers<br>• Control room operators | Supplier jobs related to the operation of the power station, for example:<br>• Service providers for the operation of power stations<br>• Service providers for the operation of coal mines |

Source: McKinsey in Bischof-Niemz and Creamer [19].

To be more accurate and specific on the employment impacts, one needs to understand the number (and type) of jobs in the energy/electricity sector in different technology value chains. The well-established coal value chain has relatively clarified characterisation of the existing and future employment—for South Africa, details can be found in the Sector Jobs Resilience Plans [20]. In comparison, the renewable energy value chains are less documented, much different and hence not directly comparable with those of coal and other fossil fuels. Fourie [21] mentions that none is available for the wind. A policy brief by the Trade and Industrial Policy Strategies (TIPS) [22] states that the units of measurement of employment chosen might partly be due to the inability to compare the various value chains fairly. Indeed, an employee is defined as any person engaged in paid work over a specified period, but other characteristics need to be compared. The same report provides an informative table with units of measurement, definitions, and their limitations, such as job-year and full-time equivalence. The smooth and successful path towards the transition involves the communities' involvement in the process [23–25]. Hence, excluding the social aspects as determinants of the transition and the communities as primary stakeholders and receivers of the impacts in the planning and modelling processes will create biased perceptions and faulty assumptions.

Studies in the literature [26–28] identify five interrelated social factors that may influence the energy transition and need to be considered from a policy and modelling perspective [29]:

- Behaviour and lifestyle: Actors in the energy transition process exhibit a variety of lifestyles and behaviours, which are also dynamic and varying in nature. These include, for example, material and non-material needs, norms, and preferences [30–32].
- Heterogeneity of actors: A diversity of actor groups is involved in the process. They differ firstly in their role in the transition (consumers or producers or policymakers), in contextual and environmental factors and their part in the dynamics of the speed of the energy transition [30,31].
- Public acceptance and opposition: Acceptance is "a favourable or positive response relating to a proposed or in situ technology or socio-technical system by members of a given social unit" [33]. Conceptually, there are three dimensions of social acceptance: socio-political, community acceptance, and market acceptance [33].
- Public participation and ownership: From the previous point, acceptance depends on public participation and ownership. This is crucial to the smooth and successful energy transition as it will allow citizens to influence and actively participate in project implementation and infrastructure development [26].
- Transformation dynamics: This aspect refers to the complexity of the transition regarding the speed of transformations and path dependencies [30,31].

The broader social aspects are misrepresented in energy models. Their quantification might provide the needed insight to bridge the modelling with the current conditions in reality. Three strategies are proposed to link energy models with the social facets: bridging (modelling and research conducted in parallel aiming at building bridges), iterating ("story and simulation" approach), and merging (in-depth integration) [29,30].

With the Sustainable Development Goals (SDGs) and Paris Agreement's Nationally Determined Contributions (NDCs) focusing on 2030, there is naturally a focus on the most effective short-term opportunities. With a rising governmental commitment to net-zero emissions objectives by 2050, there is a growing realisation of the need to examine "hard-to-abate" industries. This concept has no official definition, but it generally refers to heavy industry (cement and lime; iron and steel; petrochemicals and chemicals; aluminium), road freight transportation, and shipping and aviation. Iron and steel account for almost 9% of total energy and process $CO_2$ emissions, followed by cement (7%), chemicals and petrochemicals (5%) [34]. Without significant policy reforms, those hard-to-abate industries could account for over 35% of energy and process emissions and almost 45% of final energy use by 2050 [34].

These sub-sectors are large-scale, complex, generally located in more significant, integrated industrial settings, and frequently directly linked to infrastructures such as energy supply (power or gas), harbours, or trains for convenient and large-scale transit access energy-intensive. Production facilities are often large-scale and capital-intensive, typically rebuilt only when economies of scale justify it or major efficiency or cost gains can be realised. Most of these sectors compete in global marketplaces, making it challenging to unilaterally impose expenses associated with a national climate policy, as this would hurt the competitiveness of domestic industries. As a result, most countries have only recently begun to include these industry sectors in deliberations about climate policy, and just a handful of the NDCs include plans for emission reductions in these sectors. The best method for governments to change this is to involve critical enterprises in a discourse aimed at developing a cooperative plan for long-term decarbonisation of their activities, outlining how the various elements can be implemented and where and what kind of government assistance is required. This participation should be part of a more extensive process of just transformation.

Energy efficiency is vital in every industry. Market competition and the need to cut energy costs have motivated many large-scale process companies to focus on energy efficiency. Many older facilities, however, have room for development, notably by applying new digital control and measurement technology. According to the IEA [35], short-term efficiency enhancements might help reduce current emissions by 15–20%. Beyond implementing direct efficiency measures, enterprises can contribute to system performance in other ways. For example, fertiliser and desalination plants might use surplus electricity from variable renewable energy sources to supplement storage demands. However, the fundamental hurdle for most of these industries will be to immediately replace their reliance on fossil fuels in manufacturing and energy supply. With a net-zero aim set for 2050, a clear focus on this goal is required while evaluating options [35].

## 3. South Africa: Background

### 3.1. South Africa's Energy Transition and Its Economic Reality

South Africa's climate-related policies are extensive and include mechanisms for decarbonising the economy and facilitating new climate-resilient and transition-compatible economic opportunities. The dependence of the country's energy sector on coal makes the coal industry fundamental to the country's decarbonisation plans. The national government has committed to a scheduled retirement of coal-fired power generation to transform this sector, which will have implications for the whole coal value chain, including mining and related businesses [22]. South Africa has focused its efforts on the transition to the energy sector to mitigate climate change consequences. The transformation of the South African energy sector is not a recent phenomenon but an ongoing dynamic situation. The energy

sector has undergone four broad phases historically, according to Essex and de Groot (2019), considering the changes in provision responsibility, market structure, and access: (i) The country's electricity network in colonial South Africa (1860–1948); (ii) Electricity under the Apartheid regime (1948–1994); (iii) Electricity in the post-Apartheid area (1994–2011); (iv) Climate change, and renewable energies (2011–present).

South Africa's updated Integrated Resource Plan (IRP), published before the 2019 pandemic confirms that the country's planned energy-mix trajectory will move to a more significant share of renewables by 2030 and beyond. A quick cross-country comparison found that many other countries have initiated similar programmes to meet their environmental obligations under the Paris Agreement and SDGs framework. Eskom [36] explains how the transition is viewed in the South African context: (1) Just: Doing better for people and the planet; (2) Energy: Continued Energy Provision; and (3) Transition: Transformational change of types of fuels and business models in the energy sector.

The essential building blocks of the Just Energy Transition in South Africa are defined as (Project 90 by 2030, 2022):

(1) Accessible and Affordable electricity: International statistics suggest that up to 15% of South Africans, or around 9 million people, live without electricity. All South Africans must have access to inexpensive, low-carbon electricity to meet their fundamental needs.

(2) Corporate and Business reform: South Africa must depart from business as usual. Corporations must prioritise social and environmental challenges and implement tools to decrease emissions, pollution, and waste while still ensuring good employment.

(3) Shift in ownership of energy: Using renewable energy allows for more socially or community-owned energy generation and less corporate or privately-owned energy generation.

(4) Empowerment of workers and communities: Workers and communities should not bear the burden of transitioning to a low-carbon economy. Decent jobs and economic opportunities for all are needed to "leave no one behind".

(5) Environmental restoration and protection: Modern agricultural, mining and industrial development are degrading the quality of South Africa's soil, air, and water resources. The country must rehabilitate these areas and prevent further damage.

The prolonged slump in South African economic activity, which was heightened by the COVID-19 pandemic, combined with regulatory bottlenecks, public sector funding constraints, and reluctance in private sector investment during and in the aftermath of the state capture era has curtailed momentum with progress in the adoption of renewables, and developments are behind schedule. The COVID-19 pandemic and its associated devastation over the fragile South African economy provided the impetus for the President's Economic Reconstruction and Recovery Plan (ERRP), announced in late 2020. In the spirit of 'do not let a crisis go to waste', one of the key stimulus areas is the rapid expansion of electricity generation capacity through a diverse energy mix. This focus area has also been labelled an attempt at kickstarting a 'green economic recovery' aimed at achieving the elusive double dividend of (1) boosting economic growth coupled with associated benefits such as job creation and the reduction of inequality and (2) reducing harmful emissions, leading to environmental benefits. Barbieri et al. [37] examined the effects of green versus non-green growth trajectories. Their findings showed that green growth led to growth, emission reduction, and increased net jobs. However, the extractive sectors lost jobs as the economy reduced reliance on fossil fuels. Such results call for a better understanding of a just transition, ensuring that no one is left behind.

As stated in the literature and policy documents, social justice and consideration for potential effects on jobs and local economies must guide the timing of the shift to a low-carbon economy [38]. South Africa's high unemployment levels magnify the risks of job losses resulting from transitioning coal out of the supply mix. The choice of technology and the scale and speed of adoption of renewable energies will have consequences that vary across economic sectors and geographical regions. Mpumalanga is the province

in South Africa where most coal power plants are located. A shift away from coal will significantly affect the Mpumalanga economy, which represents around 8% of the South African economy (Figure 1), and employment levels in the mining sector in South Africa (Figure 2).

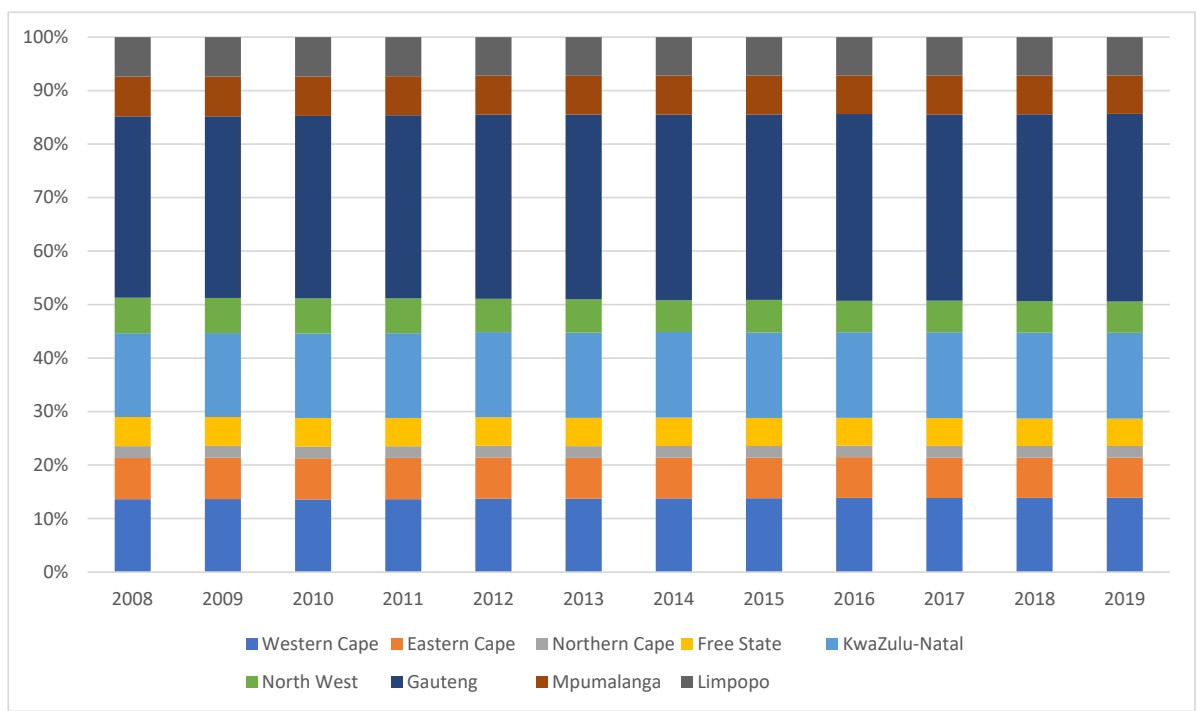

**Figure 1.** Provincial GDP constant 2010 prices. Source: StatsSA [39].

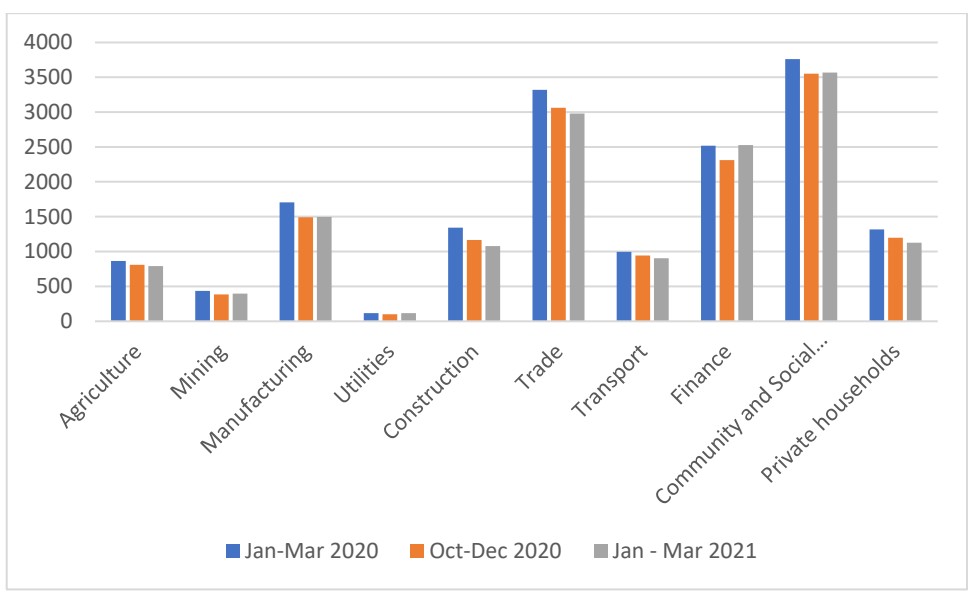

**Figure 2.** Employment by industry (in thousands). Source: StatsSA [40].

Bohlmann et al. [38] argued that the energy transition's effects are not one-dimensional among South Africa's provinces due to the distribution of coal mines and coal-fired power plants (as shown in Figure 3, which highlights how the majority of South Africa's power plants are located in one province—Mpumalanga). The impact of switching to an energy supply mix with a lower share of coal generation is dependent on other economic and policy circumstances, mainly how the global coal market responds and, as a result, how

much coal South Africa exports. Under conditions in which surplus coal resulting from lower domestic demand cannot be readily exported, the economies of coal-producing regions in South Africa, such as the Mpumalanga province, are the most severely affected. The subsequent migration of semi-skilled labour from that province to others within the country requires appropriate and timeous planning by energy policymakers and urban planners (refer to Figure 4, which shows that semi-skilled labour forms a big proportion of the South African labour force).

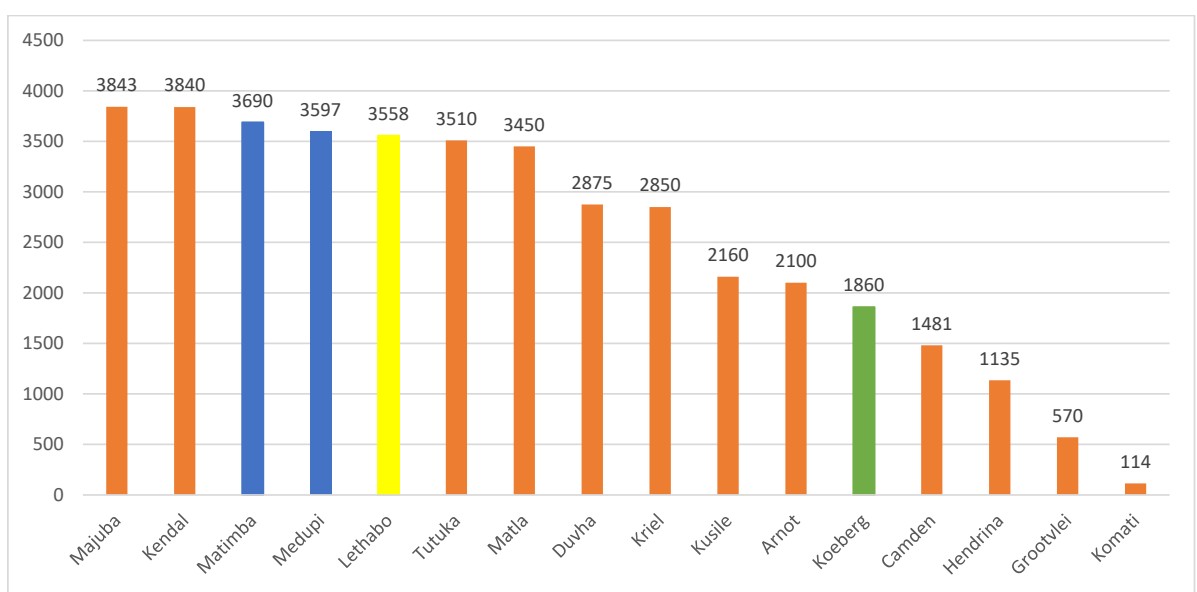

**Figure 3.** Eskom baseload 2021. Source: Eskom Integrated Report 2021 [41]. Note: Orange: Mpumalanga, Blue: Limpopo, Yellow: Free State, Green: Western Cape.

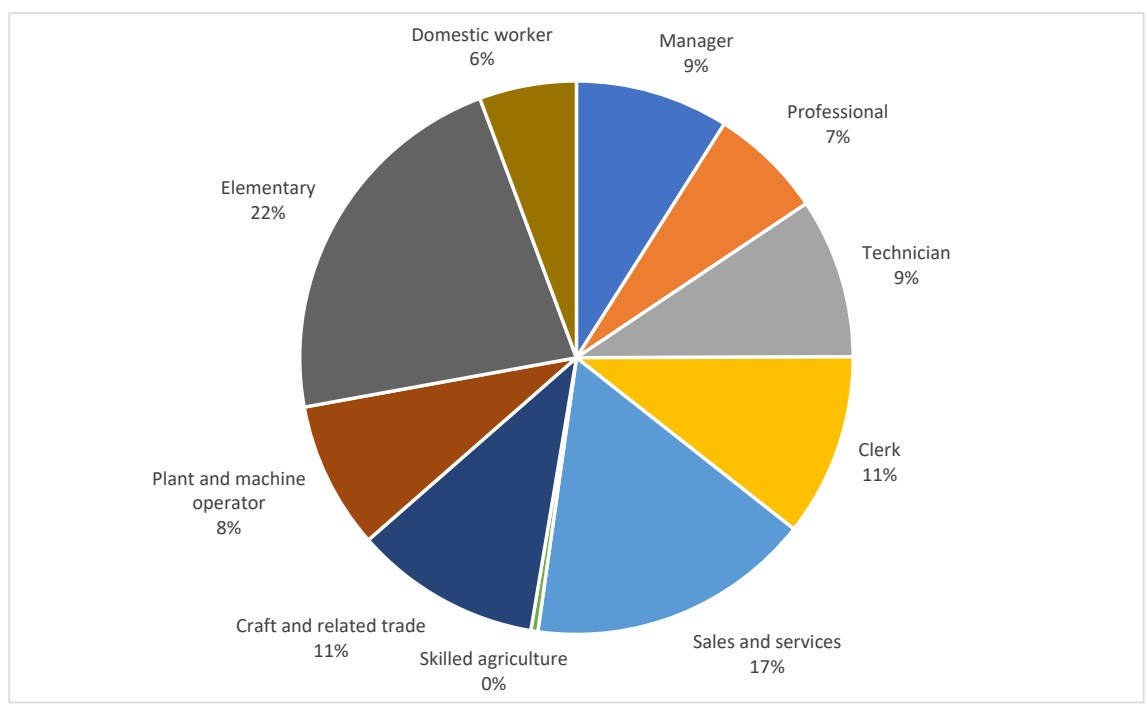

**Figure 4.** Employment by profession—April–June 2022. Source: StatsSA [40].

### 3.2. An Overview of Green Energy Instruments in South Africa

Green policies aim to promote the structural transformation of economies by reducing carbon emissions and pollution while enhancing energy and resource use efficiency [42,43]. This is of prime importance for South Africa because coal and fossil fuels (known for their high carbon intensity and environmental impacts) account for about 90% of the energy resource base or energy mix [44]. Thus, the government employs a set of energy fiscal policies, including subsidies/grants, taxes, and regulations, to promote environmental sustainability and a transition to a low-carbon economy.

- Regulation

The main policy for upscaling renewable energy installed capacity in South Africa is the Renewable Energy Independent Power Producer Procurement Programme (REIPPPP), launched in 2011 [45]. It aims to facilitate private sector investment in renewable energy projects (mainly solar and wind) through a series of competitive tender processes that award long-term contracts to private investors (also known as Independent Power Producers) for grid-connected renewable energy projects [45,46]. Since the introduction of REIPPPP, the share of renewable energy in South Africa's energy mix has increased, albeit, in relative terms, the share is still tiny. For instance, the contributions of solar and wind to total electricity generation in 2019 were 1% and 3%, respectively (IEA, 2020). In addition, the installed capacity for renewable energy increased over 10-fold within the last decade from 0.9 gigawatts in 2011 to 9.6 gigawatts in 2020 [47]. Other complementary strategies that aim to accelerate the adoption of renewable energy include the Green Transport Strategy, which was launched in 2019 to minimise the adverse impacts of the transport sector on the environment, and the Enhanced Energy Efficiency Programme, which aims to encourage South Africans to adopt a culture of energy savings.

- Taxation

Environmental taxes are essential in transitioning towards a low-carbon economy because they are meant to discourage the use of fossil fuels or fossil fuel-based products (such as plastics) and promote sustainable alternatives. In this regard, SA has introduced several environmental taxes, including the plastic bag levy (introduced in 2004/05), the electricity levy (2009/10), the incandescent light bulb levy (2009/10), the $CO_2$ tax on vehicles emissions (2010/11), the tyre levy (2016/17), and the carbon tax (which include a carbon levy on fuel and an emission tax on businesses) (2019/20) [48,49].

- Subsidies/grants

Subsidies/grants are also crucial in the suite of governments' economic levers to influence the decarbonisation of their economies, albeit at a high cost to the public purse. In South Africa, subsidies exist for hydroelectricity and solar water heating [44]. Moreover, by guaranteeing the cost of power purchase agreements for renewable energy projects via REIPPPP, the government has indirectly created subsidies for renewable energy generation [44].

### 3.3. Energy and Environmental Stylized Facts

This section provides some stylised facts about South Africa's energy sector, highlighting its reliance on coal and contribution to global $CO_2$ emissions. In 2020 over 84% of the South African population was reported to have access to electricity [50]. This is significantly higher than the sub-Saharan Africa regional average of 48% [50]. However, due to ageing coal-fired power plants, insufficient investment in power infrastructure, mismanagement of the sector, and frequent bouts of load shedding (scheduled power cuts), which started in 2008, South Africa's electric power sector has struggled to provide adequate and reliable power to its end users [48]. The lack of reliable power supply has affected the country's industries and economic growth.

As a result of the country's abundant coal reserves and consistent domestic coal production, South Africa predominantly uses coal-fired power generation to meet its

electricity generation needs. Fossil fuel-derived generation accounted for almost 80% of the total energy supply and around 80% of electricity generated in South Africa in 2019. This has been consistent since the 1990s [2,51,52] (see Figures 5 and 6).

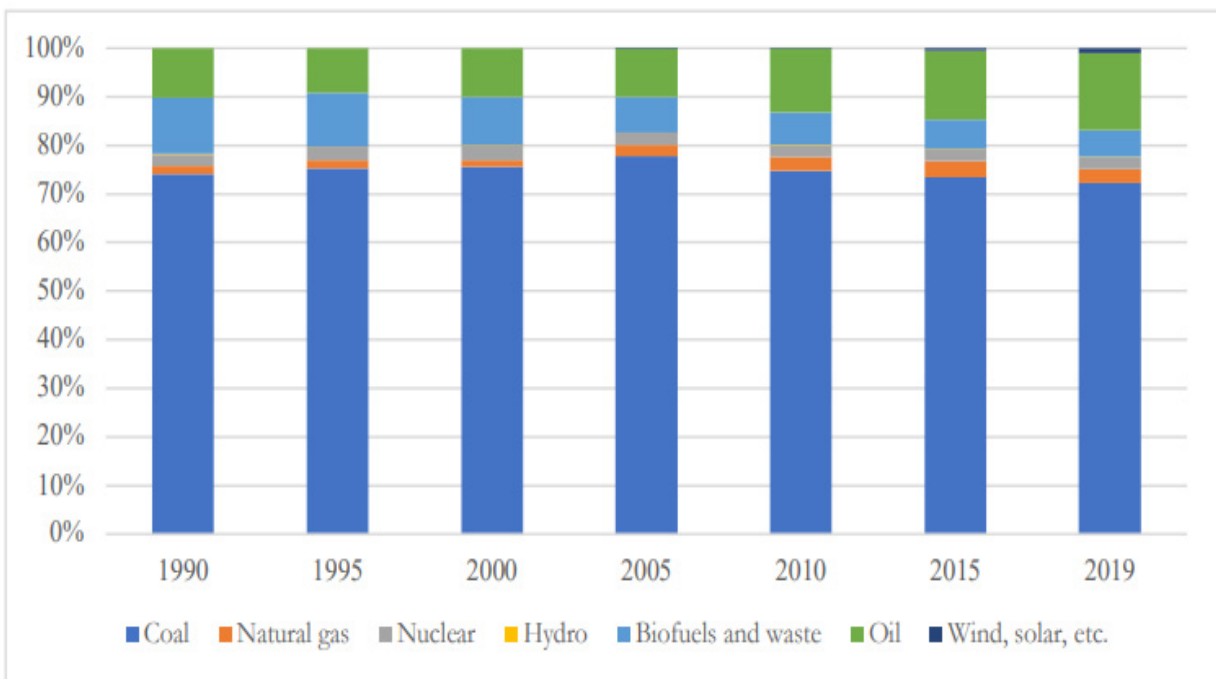

**Figure 5.** Total energy supply by source (in TJ). Source: IEA [51].

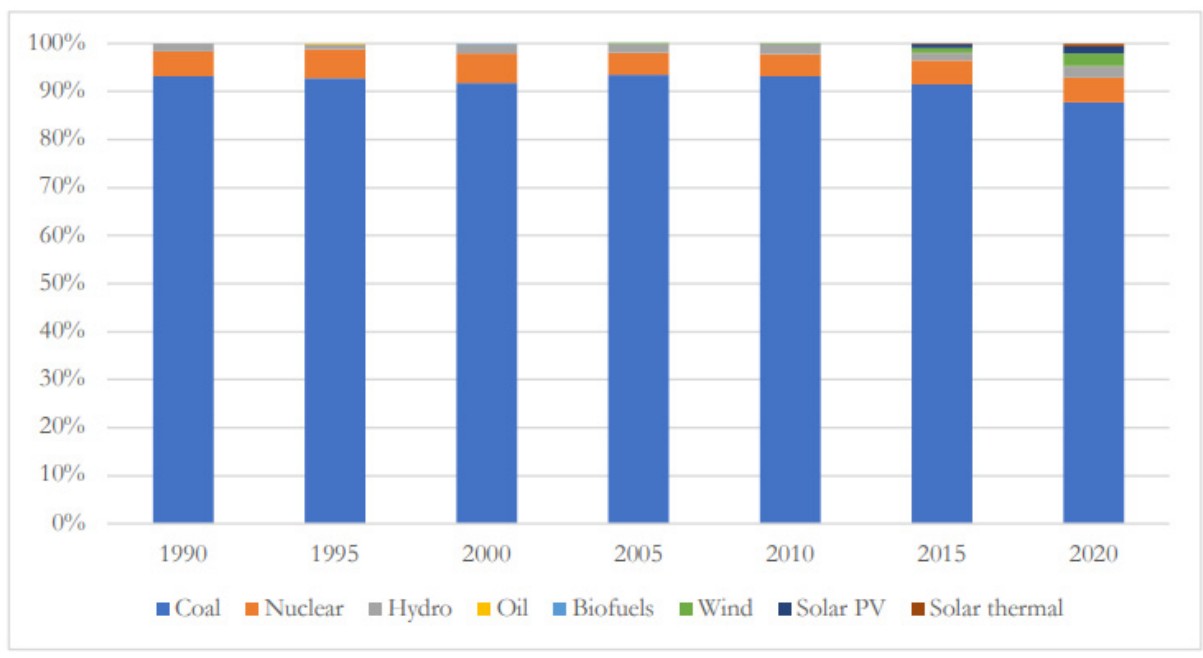

**Figure 6.** Electricity generation by source (GWh). Source: IEA [51].

South Africa is the 14th largest emitter of greenhouse gases (GHGs) in the world. Its $CO_2$ emissions are principally due to a heavy reliance on coal in the electricity generation sector (Figure 7).

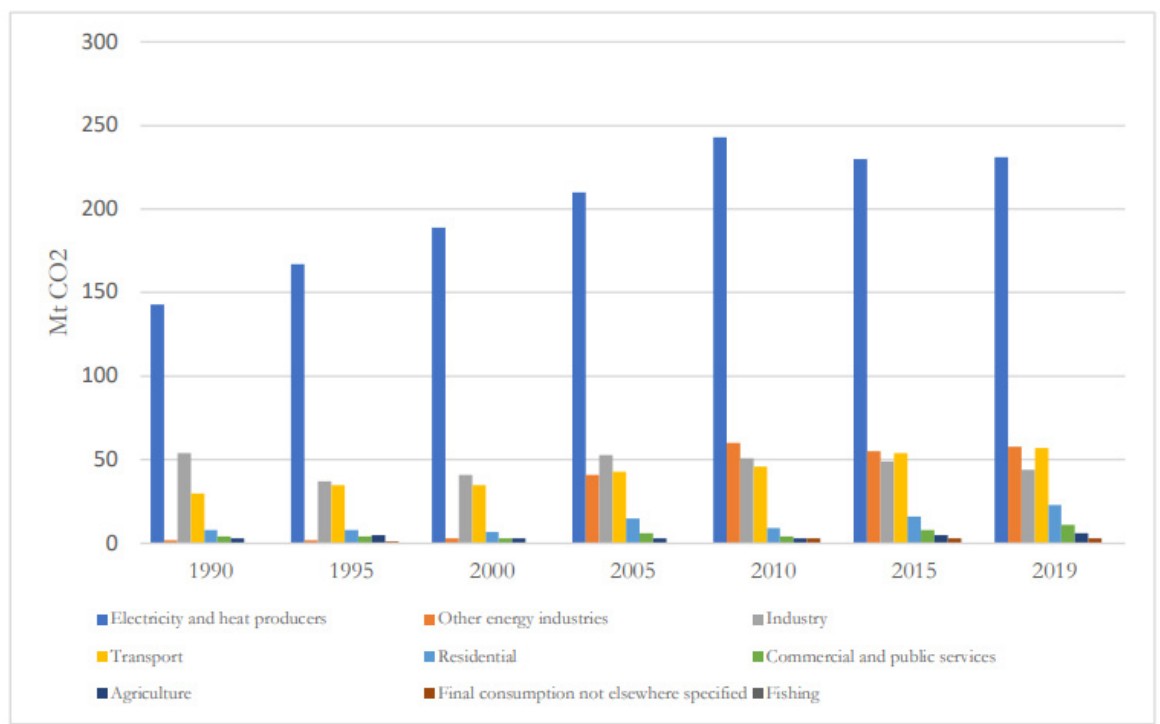

**Figure 7.** CO$_2$ emissions by sector (Mt CO$_2$). Source: IEA [51].

Globally, the share of renewable energy in final energy consumption has been increasing since the implementation of the Paris Agreement, with modern renewable sources (excluding traditional uses of biomass) growing at a faster pace than global energy consumption, which has allowed the share of modern renewables in total final energy consumption to increase marginally to 11.5% in 2019, from 11.1% in 2018 [53]. In the South African case, the share of modern renewables in total final energy consumption is following an upward trend, increasing to 10.5% in 2019 from 10.19% in 2018 (Figure 8) [53]. However, this trend is not enough to reach the Net Zero Emissions by 2050 Scenario by the EIA, which is in line with the United Nations Sustainable Development Goal.

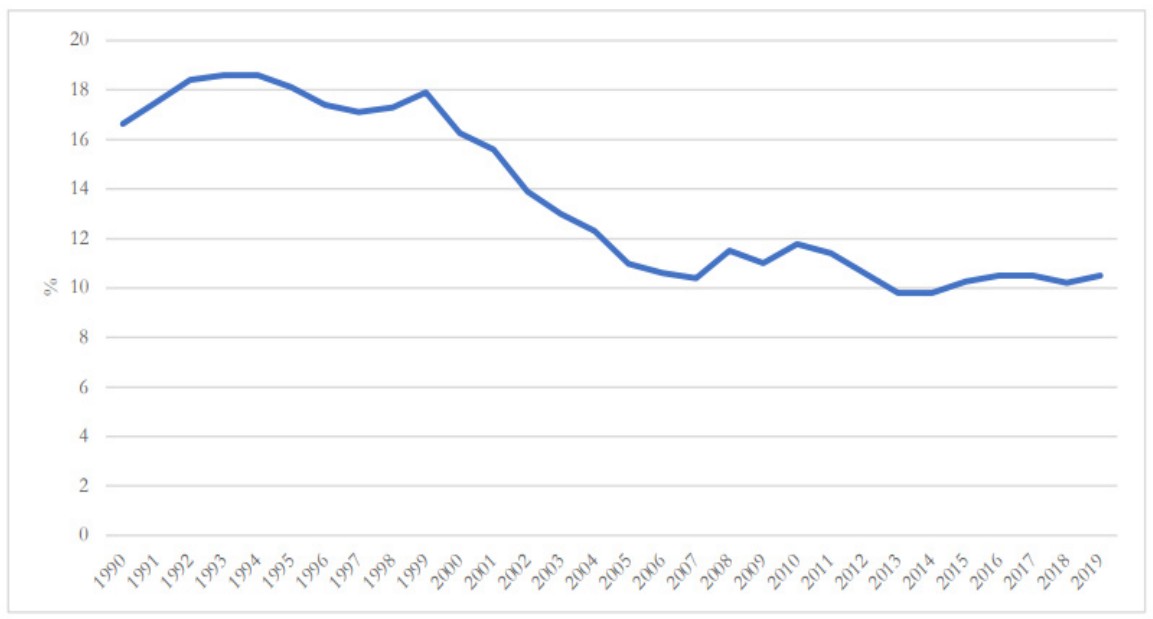

**Figure 8.** Renewable share in final energy consumption (SDG 7.2). Source: IEA [53].

## 4. Materials and Methods

### 4.1. Computable General Equilibrium (CGE) Models

The methodology used in this project employs a regional-dynamic computable general equilibrium (CGE) model. CGE models are multi-sector macroeconomic policy simulation models that can project the economy-wide effects of various external shocks and/or policy interventions. With specific reference to this project, dynamic CGE models have the added advantage of producing more insightful analyses on the adjustment path of any long-term energy planning scenario.

CGE models highlight the various economic interlinkages, account for price-induced behaviour, and the effects of resource constraints. Thus, CGE models are well suited for determining the economy-wide impact of a specific shift or change in a particular micro or macroeconomic variable. Globally, CGE modelling has become one of the preferred methodologies for practical policy analysis. The underlying database of a CGE model is based on a representative reference year for the specific economy. Large-scale datasets such as supply-use tables (SUT) and/or social accounting matrices (SAM) are typically used to build the core CGE model database. One of the critical features of the computational framework of modern CGE models is its ability to cope with many highly disaggregated dimensions in an integrated framework. Therefore, modellers can conduct simulations across multiple industries, commodities, occupations, household types, and regions and be assured that all general equilibrium balancing conditions are accounted for. Credibility-enhancing detail, such as the disaggregation of the final purchaser's prices into the basic price, margin costs, and tax components, are added within this framework. To isolate and measure the economy-wide impact of any proposed policy change or shock, the CGE modeller first establishes a business-as-usual (BAU) baseline forecast of the economy without the exogenous policy change or shock being considered. Secondly, a simulation is performed where the exogenous policy shock (the evaluated policy) is imposed on the model. Results are typically reported as percentage change deviations between the first 'baseline' simulation run and the second 'policy' simulation run. Therefore, these results will indicate the impact of the specific policy being studied.

This study will use the University of Pretoria General Equilibrium Models (UPGEM) suite, based on the MONASH and TERM models developed by the Centre of Policy Studies and documented in Dixon et al. [54] and Horridge et al. [55]. Our application of UPGEM combines a regional-dynamic CGE model of the South African economy similar to that described in Roos et al. [56] and an energy-environmental version linked to an external emissions database identical to that described in Van Heerden et al. [57]. The latter version includes an energy and emissions accounting model, different equations that will enable inter-fuel substitution in electricity generation and different mechanisms that allow for the evaluation of the environmental impacts caused by changes in the energy generation mix of the country in combination with the substitution of energy inputs in the transport and mining sectors. The emissions and energy data methods used to develop the emissions database in the model are based on Blignaut et al. [58] and Seymore et al. [59], who developed energy inventories for South Africa. For the UPGEM base year in this paper, its core and emissions database has been updated and calibrated to 2017 data, following the methods described in Roos et al. [56].

### 4.2. Simulation Design

As is standard for dynamic CGE models, we first simulate a business-as-usual (BAU) baseline run that reflects the projected evolution of the economy in the absence of any policy interventions. The policy run, capturing the various policy scenarios and shocks related to the energy transition, is then simulated and contrasted against the BAU baseline run.

For the baseline run, we use macroeconomic projections available in the latest Budget Review from National Treasury. Alternative projections from international institutions such as the World Bank or International Monetary Fund are virtually the same, with only minor

differences in some macro variables, but all reflecting similar trends [50]. The low baseline growth projected in the medium term of below 2% per annum is extended through the simulation horizon of 15 years, increasing only slightly to an average of 2.5% per annum beyond 2025. However, since the focus of the work and methodology is to determine how the various policy shocks will cause a deviation away from the baseline—that is, to isolate and measure the impact of the shock—and not the exact path of the baseline itself, the results produced remain credible even if the baseline projections do not exactly materialise as modelled.

Four policy simulations are designed and run as part of this project. Both global and domestic events inform shocks, trends, and policy preferences indicated by the government. The first policy shocks (S1A and S1B) capture the effects of the increasing trend seen in rising costs related to financing investment for fossil fuel projects. With specific reference to South Africa and the implementation of this shock in UPGEM, this translates into a decrease in investment in the coal and refined petroleum industries for any given rate of return as financing costs and associated risks rise. As a second part of this simulation, we do the same for the refined petroleum industry, given its strong linkages to the coal industry in South Africa.

The second policy shock (S2) extends the first by capturing possible spillover effects into the broader economy of deteriorating financing, fiscal, and risk conditions. Within UPGEM, this is modelled as an economy-wide increase in the required rate of return of investors across all industries for any given amount of investment or less investment at any offered rate of return than before.

The third policy shock (S3) captures the implementation of a hypothetical, yet increasingly likely, restriction on coal export in the future. This is modelled as a reduction or inward shift of the export demand curve for coal from South Africa. Combined, the first three policy simulations aim to provide a picture of possible shocks to the economy in future in the absence of an energy transition. Policymakers must therefore consider the likelihood that financing for fossil fuel investments will be harder to obtain, broader fiscal risks may emerge, and our local coal industry will come under increasing pressure in the future to find buyers for its output.

The fourth policy shock (S4) relates specifically to the impacts of the required energy transition in South Africa. It captures the regional and macroeconomic effects of the proposed change in the country's electricity generation mix. This is modelled as a technological change away from the use of coal towards non-coal inputs in electricity production following the modelling strategy used by Bohlmann et al. [38].

Given the nature of the research and uncertainty about the timing and magnitude of these shocks, the policy shocks are not calibrated to any detailed, proposed intervention. Still, they should instead be viewed in a 'what-if' benchmark sense. This strategy allows policymakers to use their judgement about the likely size and timing of the potential shocks, considering other policy elements exogenous to UPGEM, and use the benchmark analysis provided by these simulations to determine what the probable economy-wide effects will be. Similarly, the year in which the shock is first imagined to occur should be interpreted as year t when looking at the results of the policy simulations.

## 5. Results

Scenario 1A (S1A) simulates a benchmark 1%-point increase in financing costs in years t and t + 1 related to coal industry investment projects. For any given rate of return, investors are willing to provide less capital than in the base run due to the deterioration in risk and financing conditions. This scenario simulates a shift in risk-adjusted preferences by financiers and investors away from new coal projects.

The results that follow in Figures 9–12 indicate the damage to the coal industry, which also disproportionately affects the Mpumalanga region and its economy due to the drop-in investment in the sector. Nationally, the damage to the coal industry, and by extension,

the Mpumalanga economy, is largely offset by improved performance in other sectors as resources are freed up and labour is allowed to move over time.

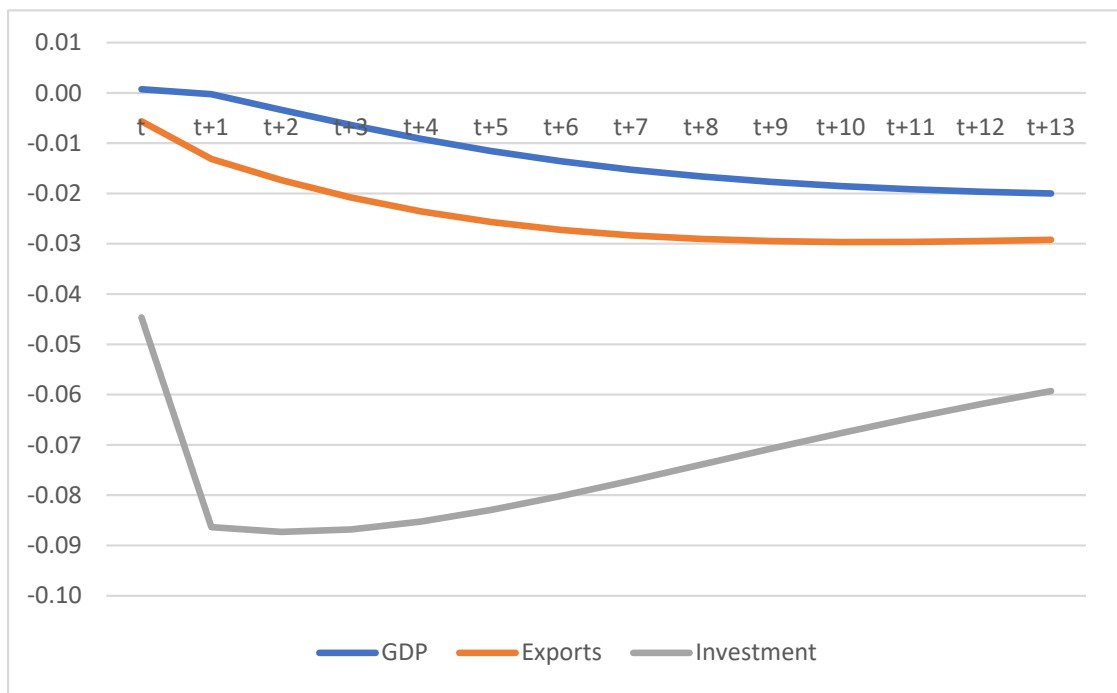

**Figure 9.** Key national macroeconomic results for S1A.

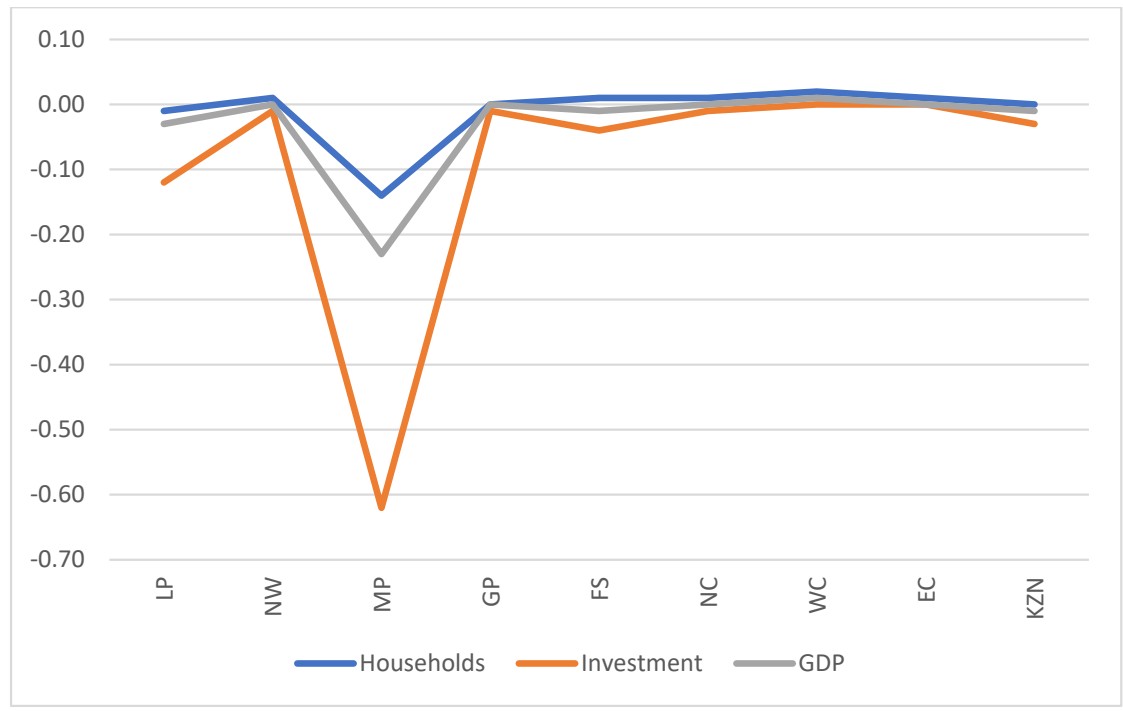

**Figure 10.** Key regional macroeconomic results for S1A (long-run).

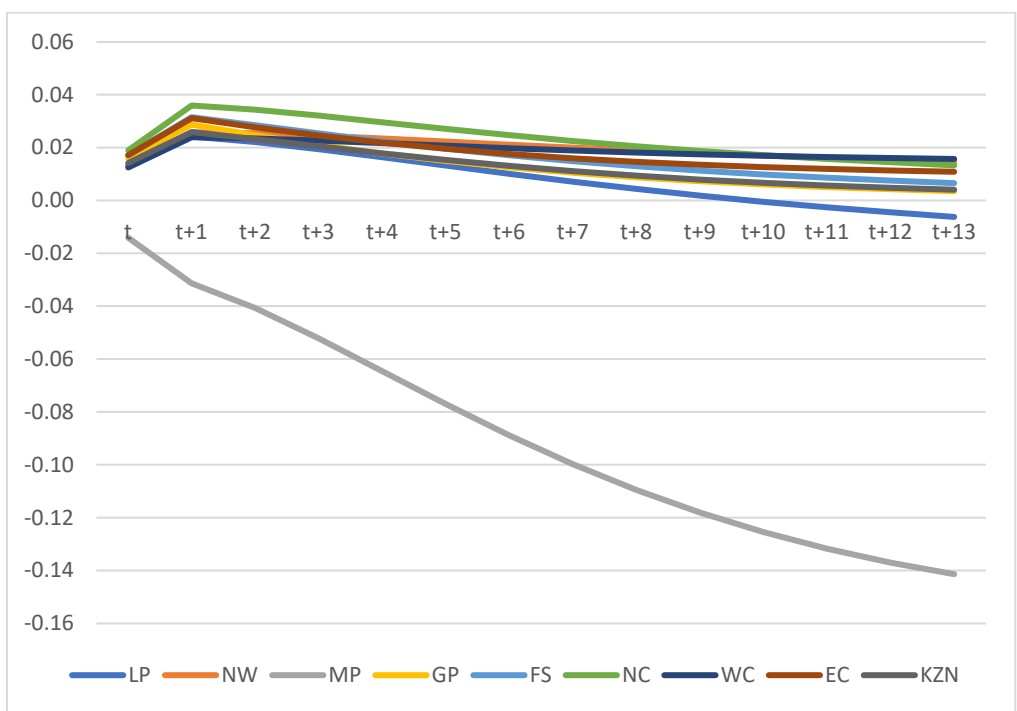

**Figure 11.** Household consumption results in selected regions for S1A.

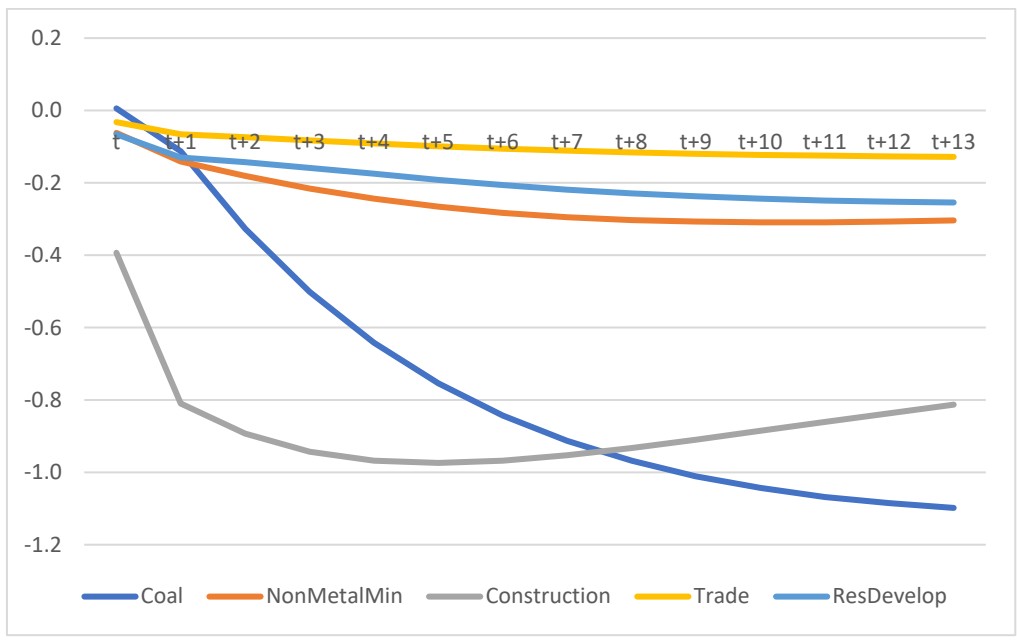

**Figure 12.** Selected industry output results in Mpumalanga region for S1A.

Scenario 1B (S1B) replicates S1A for the refined petroleum industry in South Africa, given its strong linkages to the coal industry. At a firm level, the most notable example is Sasol, which converts coal into synthetic fuels and chemicals. This simulation aims to capture the impacts of reduced investor sentiment in sectors with close ties to the fossil fuel industry, as has already started to happen in special cases. The results in Figures 13 and 14 again show that the Mpumalanga region is very exposed to such a change, with the refined petroleum sector being most directly impacted, and the coal industry being negatively impacted. Should such a policy change be broadly implemented by financiers, a combination of S1A and S1B may emerge over time.

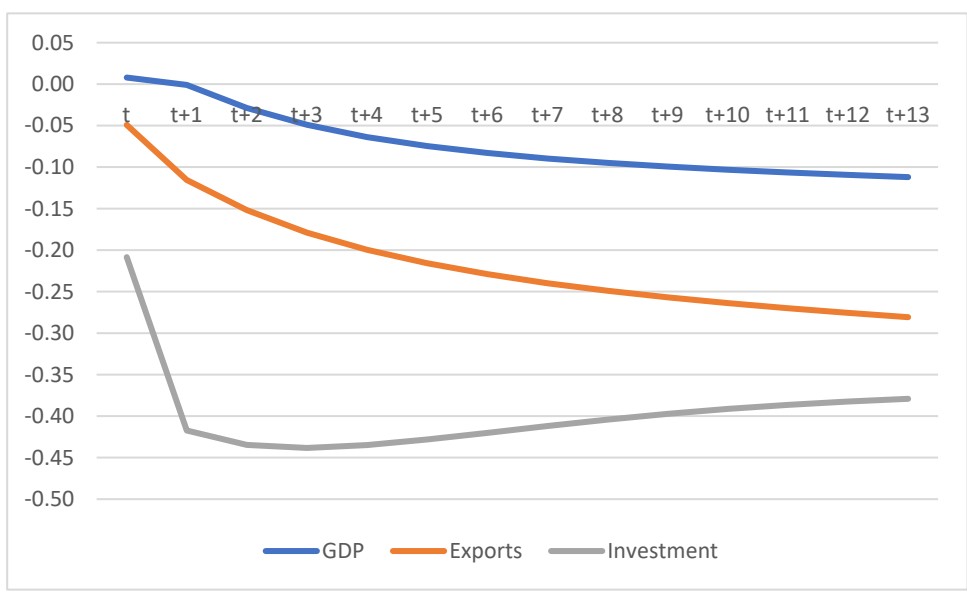

**Figure 13.** Key national macroeconomic results for S1B.

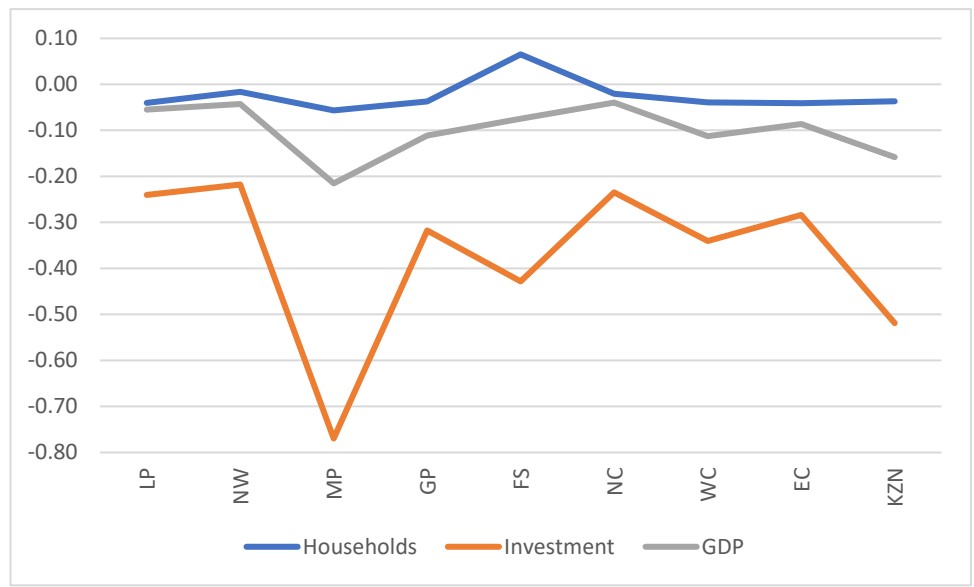

**Figure 14.** Key regional macroeconomic results for S1B (long-run).

Scenario 2 (S2) simulates a benchmark 0.5%-point increase in economy-wide financing costs in years t and t + 1. For any given rate of return, investors are willing to provide less capital than in the base run due to the general deterioration in risk and financing conditions. The results that follow in Figures 15 and 16 indicate the damage to the general macro economy due to the drop in investment across all industries. GDP falls by over 1%, with aggregate investment spending down over 9%. This significantly reduces the country's ability to build much-needed capital and raise productivity. Scenario S2 is designed to capture the potential spillover effects that may arise from adverse economic conditions related to the increased financing costs of fossil fuel projects, including those potential projects beyond the current Kusile and Medupi, as per the IRP. This type of simulation is also a proxy for various other investment-related shocks to the economy, including a general loss of business and investor confidence, stemming from factors related to weak institutions and policy choices incompatible with global commitments that may affect South Africa's ability to raise funding.

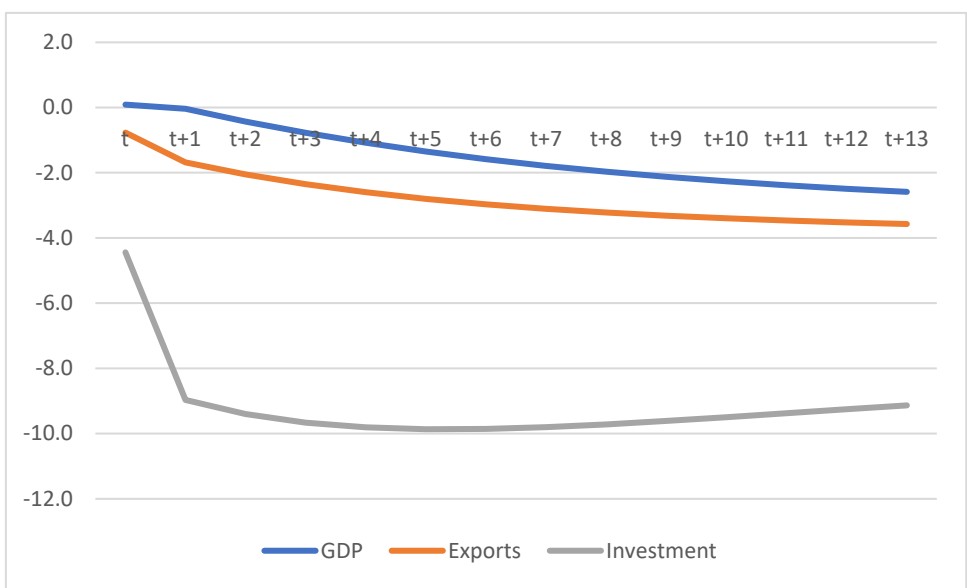

**Figure 15.** Key national macroeconomic results for S2.

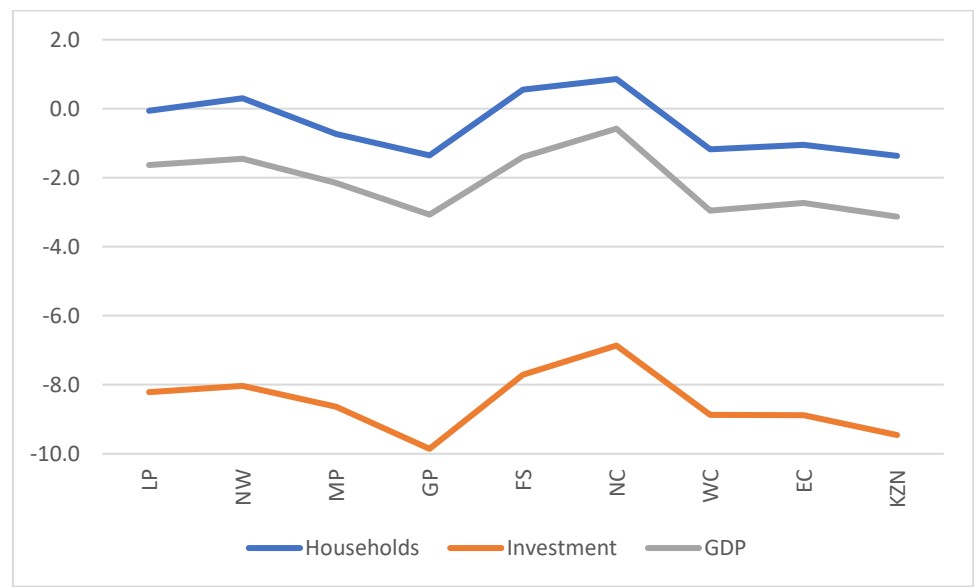

**Figure 16.** Key regional macroeconomic results for S2 (long-run).

Scenario 3 (S3) simulates a gradual and partial embargo on coal exports in which export demand is reduced by 50% over three years. Although this scenario is not likely to materialise in the short run, there is a high probability, given increasing commitments to reduce fossil fuel use globally in the future, that such a shock may eventually impact South Africa's coal industry. In this case, year t should be interpreted as the future period in which such a policy shock becomes likely.

The results that follow in Figures 17–21 highlight the reliance on strong export demand for the local coal industry and Mpumalanga's reliance on a strong-performing coal industry. Once again, the winners and losers are industry and region-specific. Nationally, and on aggregate, the economy shows only some damage in the short run, with positive outcomes in most macro variables in the medium to long run. Aggregate exports, perhaps surprisingly to many, show a slight gain relative to the baseline in a reverse case of the so-called 'Dutch disease'. Whereas most macro variables and regions show hardly any damage from a coal-export embargo, the clear losers are those directly tied to the policy shocks, which, of

course, involve the coal industry and, by implication, Mpumalanga, which is where the vast majority of coal resources are sourced from.

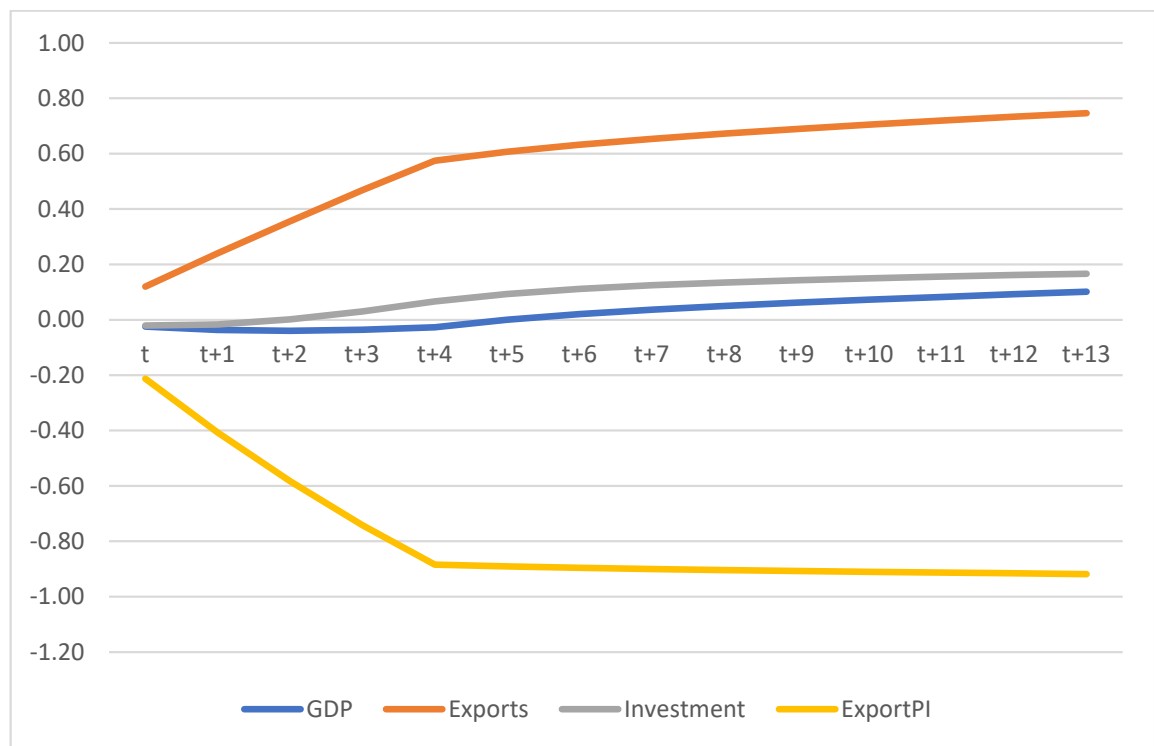

**Figure 17.** Key national macroeconomic results for S3.

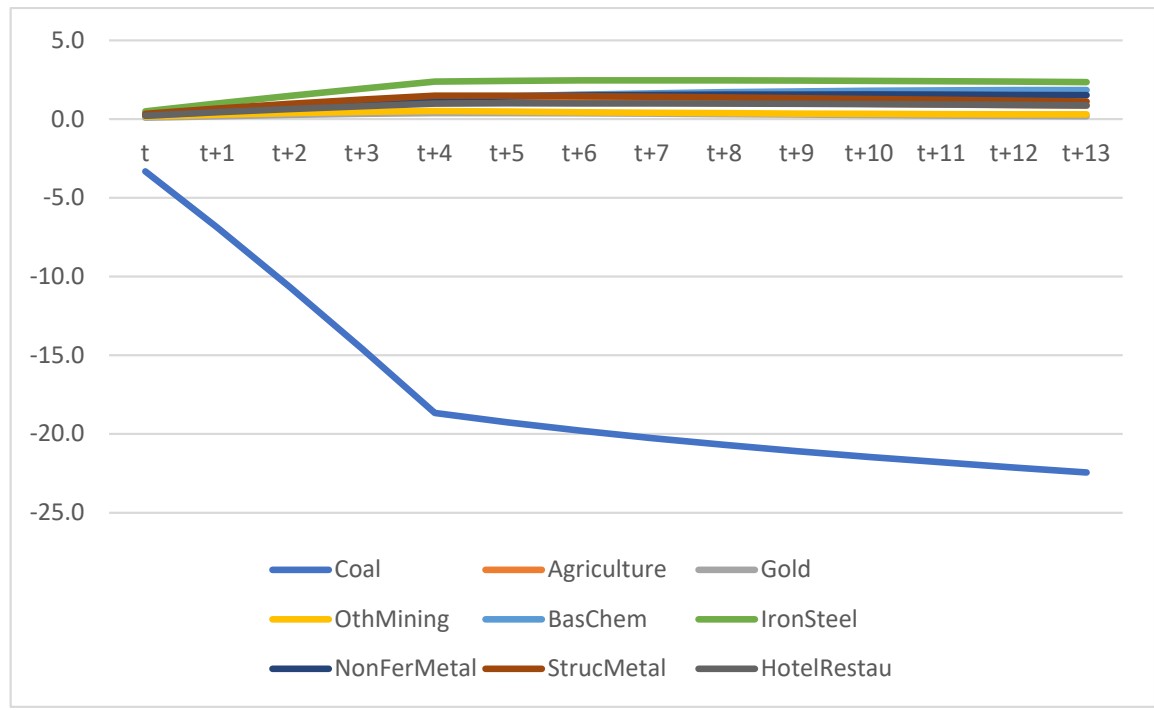

**Figure 18.** Key national export results for S3.

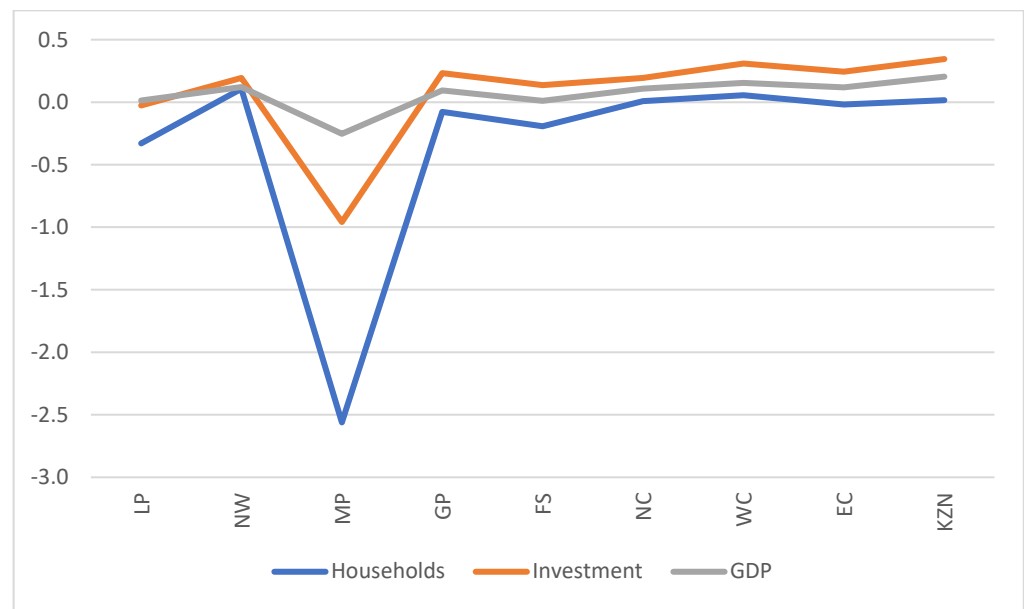

**Figure 19.** Key regional macroeconomic results for S3 (long-run).

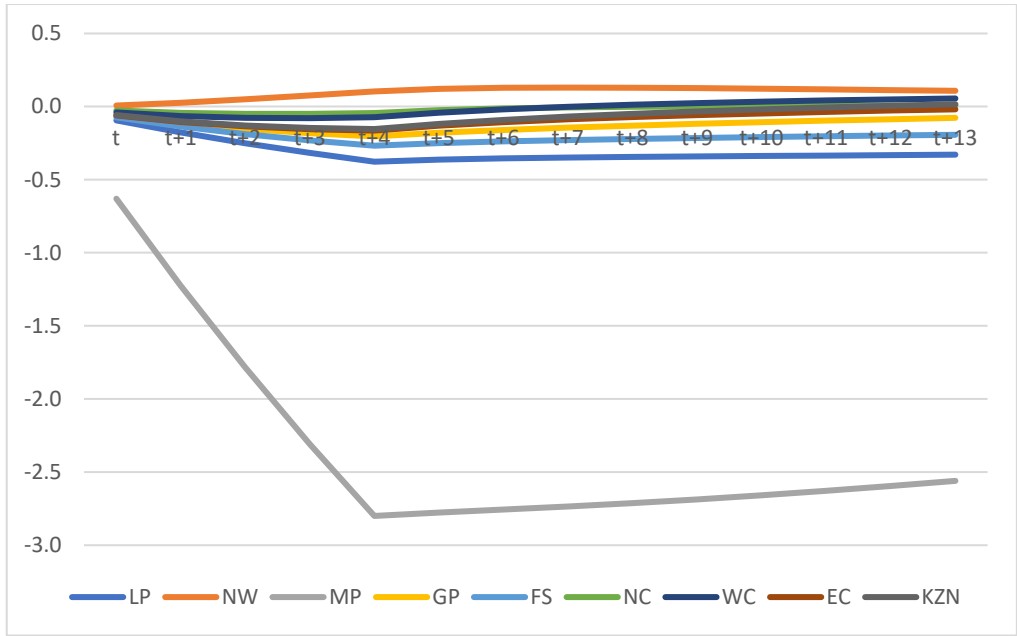

**Figure 20.** Household consumption results in selected regions for S3.

Scenario 4 (S4) simulates a change in the input structure of electricity generation in South Africa win line with the broader energy transition envisioned in the IRP and through its SDG commitments. Specifically, we model a benchmark technological shift in electricity production away from coal use (50% less over a two-year period) towards non-coal inputs. Realistically, such a transition will likely occur over many years, but this simulation deliberately models a quick transition as part of the benchmark modelling strategy. This particular scenario builds on the work done by Bohlmann et al. [38] by using a dynamic version of the regional UPGEM instead of a static long-run version.

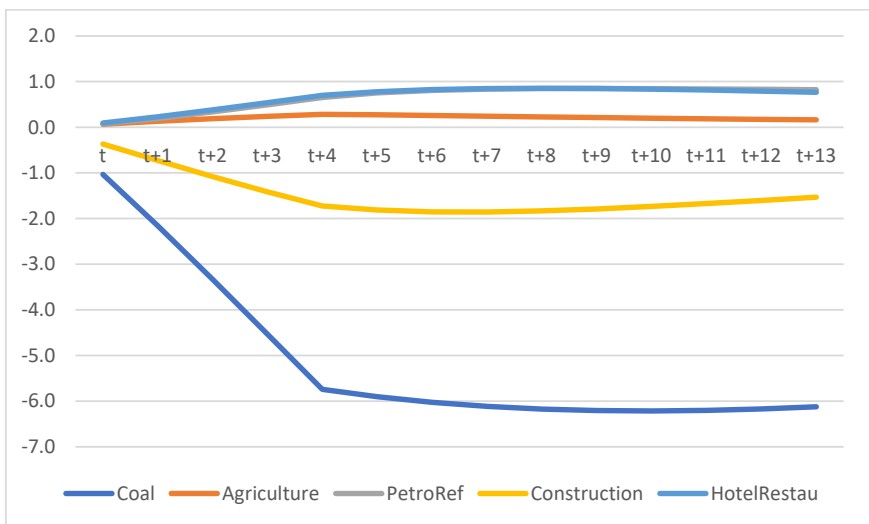

**Figure 21.** Selected industry output results in Mpumalanga region for S3.

The results that follow in Figures 22–26 again highlight the real threat of the energy transition to the Mpumalanga economy, yet also indicate that from a national perspective, there is very little to worry about should the necessary adjustment mechanisms and labour movements be facilitated. After an initial adjustment period, the national GDP grows above baseline levels due to substantial investment and export growth. Predictably, the Mpumalanga economy falls well below baseline performance, but it should also be noted that no targeted mitigation measures, as planned, are modelled in this scenario. It is further remembered that the damage to the coal industry, and as a result, the Mpumalanga economy, is necessary by design as per our global SDG and NDC commitments. The local coal industry has benefited from our reliance on it to power our economy for more than the last century. Still, the time has come to move to cleaner and more sustainable sources of electricity generation. As indicated in the results from S1 to S3, by ignoring or postponing the need to transition too long, the Mpumalanga economy will get left behind regardless of local policy choices as financing and export conditions are likely to increase harder.

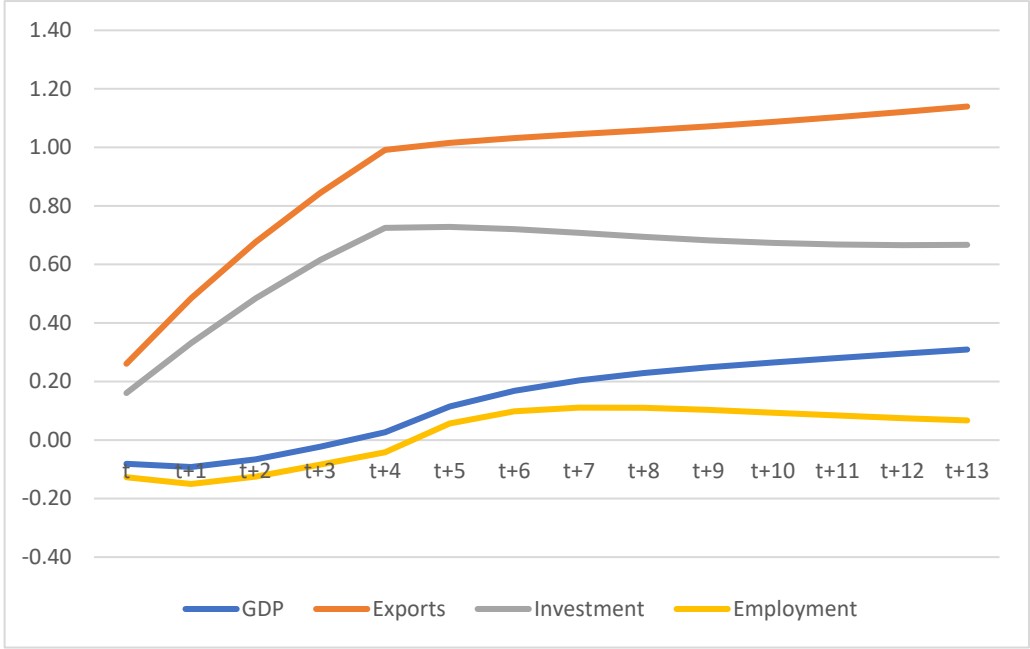

**Figure 22.** Key national macroeconomic results for S4.

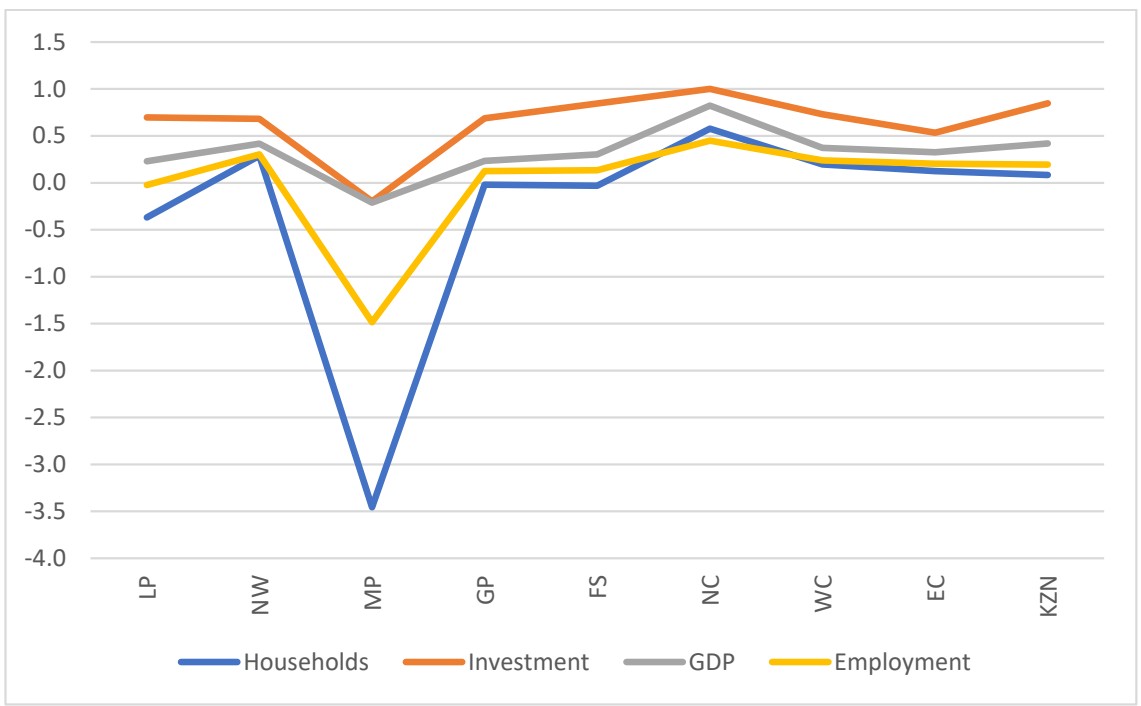

**Figure 23.** Key regional macroeconomic results for S4 (long-run).

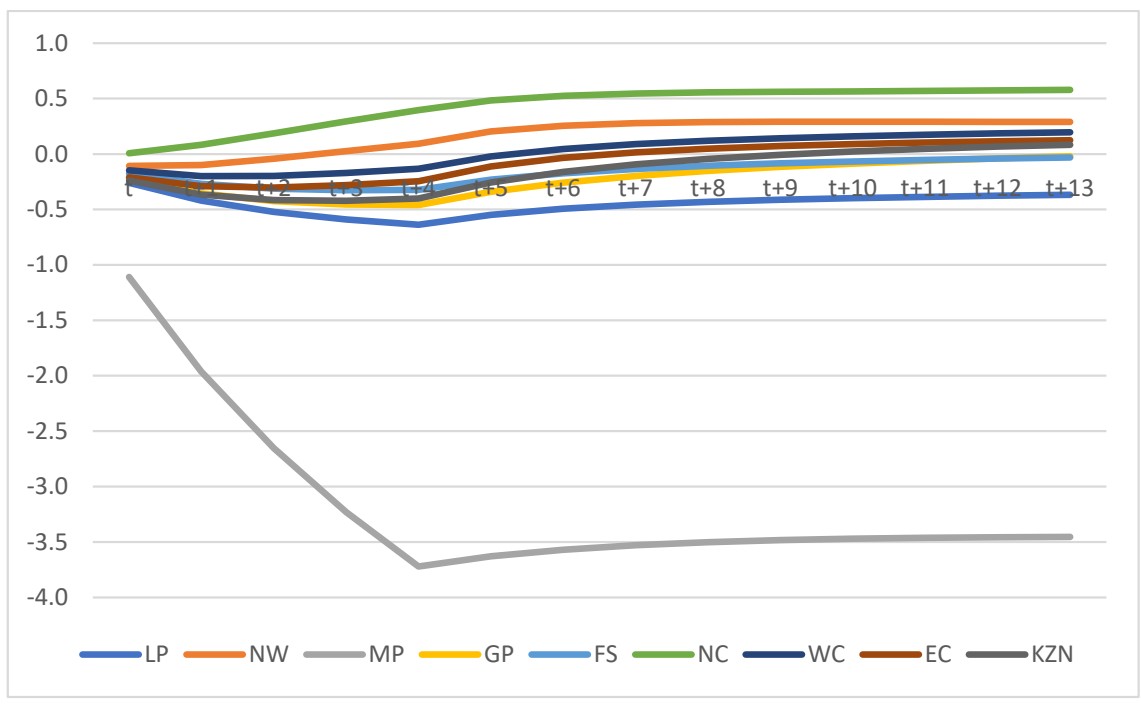

**Figure 24.** Household consumption results in selected regions for S4.

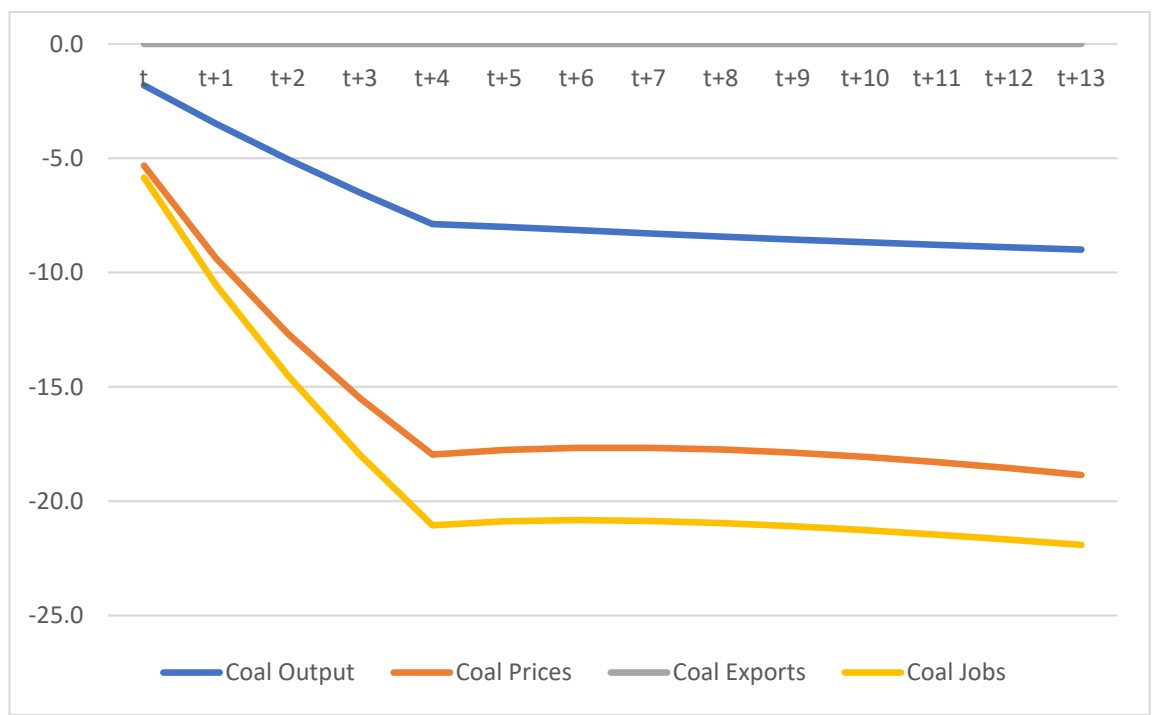

**Figure 25.** National coal industry results for S4.

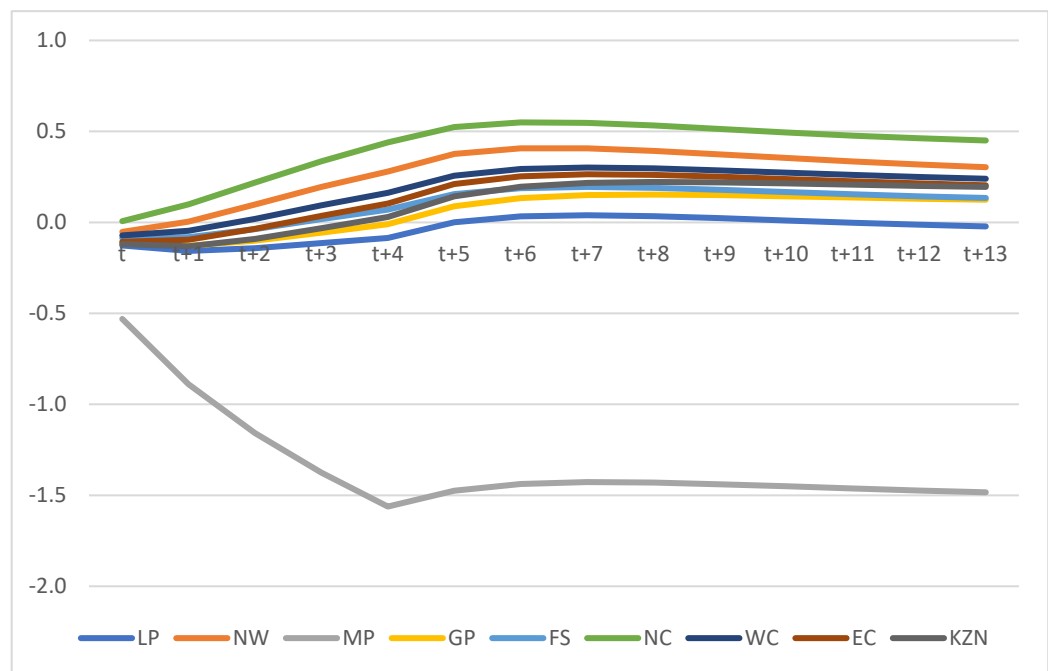

**Figure 26.** Employment results in selected regions for S4.

## 6. Conclusions and Policy Implications

This study presents the empirical simulation results of four policy shocks on South Africa's economy as part of the energy transition. The simulation design starts with the business-as-usual (BAU) baseline run reflecting the projected evolution of the economy in the absence of any policy interventions. The four policy simulations are designed to capture the effects of increasing costs related to financing investment for fossil fuel projects, the possible spillover effects of deteriorating financing, fiscal, and risk conditions, hypothetical

restrictions on coal export in the future, and the impacts of the required energy transition in South Africa.

The results show that the Mpumalanga region, which is a significant producer of coal, is highly exposed to policy changes related to the fossil fuel industry. Scenarios 1A and 1B, which simulate a benchmark increase in financing costs related to the coal and refined petroleum industries, respectively, show damage to the coal industry, negatively impacting the Mpumalanga economy. Scenario 2, which simulates a benchmark increase in economy-wide financing costs, indicates damage to the general macro economy, but the negative impacts on coal production and employment are mitigated by improved performance in other sectors over time. Scenario 3, which simulates a hypothetical restriction on coal export, highlights the potential for the local coal industry to come under increasing pressure in the future to find buyers for its output. Scenario 4, which simulates the impacts of the required energy transition in South Africa, suggests that transitioning away from coal towards greener electricity production can significantly positively impact the environment and public health while generating employment and income growth in the economy.

This study provides useful insights into the possible shocks to the economy in the absence of an energy transition, and policymakers can use the analysis provided by these simulations to determine the probable economy-wide effects of policy interventions. Based on the simulation results, policy recommendations can be made to manage the energy transition in South Africa.

Firstly, policymakers must consider the high exposure of the Mpumalanga region to policy changes related to the fossil fuel industry. To mitigate the negative impacts of the energy transition on the region, policymakers must implement deliberate strategies to support the transition of workers and communities from coal-dependent industries to alternative employment opportunities. This can include investment in infrastructure, re-skilling, and education programs. We believe this must be intentional.

Secondly, to address the potential for the local coal industry to come under increasing pressure to find buyers for its output in the future, policymakers must explore options for diversifying the economy away from coal with some urgency. This can include investment in renewable energy technologies, promoting energy efficiency, and supporting the growth of sustainable industries.

Thirdly, to facilitate the energy transition in South Africa, policymakers must provide a stable policy environment that encourages investment in renewable energy and energy efficiency technologies. This can include long-term renewable energy targets, transparent regulatory frameworks, and policies that support the deployment of renewable energy technologies, such as feed-in tariffs and tax incentives.

Finally, policymakers must consider the potential spillover effects of deteriorating financing, fiscal, and risk conditions. To mitigate these negative impacts, policymakers must implement strategies that promote macroeconomic stability, such as sound fiscal and monetary policies, and address systemic risks that may arise from the energy transition.

The simulation results include policy suggestions for successfully managing South Africa's energy transition. These suggestions stress the importance of assisting workers and communities in the severely exposed Mpumalanga region by investing in infrastructure, retraining, and education initiatives. To address the issues that the local coal sector will face in the future, it is also suggested that the economy diversify away from coal. This can be done by making investments in renewable energy technology, promoting energy efficiency, and supporting sustainable enterprises. By setting long-term goals, clear laws, and supportive policies, governments may create a stable climate that promotes investments in renewable energy and energy efficiency. Last but not least, macroeconomic stability measures should be used to adopt efforts to reduce harmful spill over effects, such as deteriorating risk and financing circumstances. With everyone benefiting from the environmental, health, economic, and financial advantages of switching to renewable energy sources, these proposals are meant to serve as a roadmap for policymakers as they work to achieve a just and sustainable energy transition in South Africa.

In conclusion, the simulation results (See summary Table 2) highlight the importance of managing the energy transition in South Africa carefully. While transitioning away from coal towards non-coal inputs in electricity production can have significant positive impacts on the environment and public health while generating employment and income growth in the renewable energy sector, policymakers must address the potential negative impacts on regions and industries that are highly dependent on the fossil fuel industry. The policy recommendations provided in this study can inform policymakers on strategies to support a just and sustainable energy transition in South Africa, which leaves no-one behind.

**Table 2.** Key findings.

| Scenario | Key Findings |
|---|---|
| Scenario 1A | <ul><li>The coal industry in Mpumalanga will be significantly impacted by tougher investment conditions</li><li>Regional effects dominate as nationally, the impact is minimal over the medium to longer-term</li><li>Investment in fossil fuel projects has already come under scrutiny, which suggests that this type of scenario will happen and may already be happening, making the energy transition inevitable</li></ul> |
| Scenario 1B | <ul><li>Similar to S1A, industries closely tied to fossil fuels in their value chain that will come under greater scrutiny from ESG investor groups will be significantly impacted</li><li>Mpumalanga and the coal industry will be further exposed if harsher investment conditions extend to the refined petroleum sector in South Africa</li></ul> |
| Scenario 2 | <ul><li>An economy-wide deterioration in investment, or capital supply, is extremely harmful to the economy and its ability to grow its levels of capital per worker</li><li>South Africa's weak fiscal position in recent years and series of shocks to investor confidence during the period of state capture, suggests that shocks of this nature may have occurred on multiple occasions already, explaining the low levels of GDP and capital per worker growth.</li></ul> |
| Scenario 3 | <ul><li>An economy-wide deterioration in investment, or capital supply, is extremely harmful to the economy and its ability to grow its levels of capital per worker</li><li>South Africa's weak fiscal position in recent years and series of shocks to investor confidence during the period of state capture, suggests that a shock of this nature may have occurred on multiple occasions already, explaining the low levels of GDP and capital per worker growth.</li></ul> |
| Scenario 4 | <ul><li>The electricity generation technology shift away from coal in coming years will reduce the size of the Mpumalanga economy given the concentration of the coal industry in the region.</li><li>If the domestic use of coal is reduced due to the technology shift, the industry will not be able to export excess supply of coal beyond the short run as the shift away from fossil fuels is expected to become a global trend in the medium to long run.</li><li>The ability of displaced coal mining workers from Mpumalanga to move into other jobs in other regions will ultimately determine the severity of the impact on jobs.</li><li>Mitigation strategies focussed on retraining workers and repurposing existing coal mines and coal-fired power station sites in Mpumalanga for renewable generation sources and take advantage of the well-developed transmission network in the region will be crucial in limiting the disruption to the local economy and its workforce.</li></ul> |

**Author Contributions:** Conceptualization, R.I.-L.; Methodology, H.R.B.; Software, H.R.B.; Formal analysis, H.R.B., J.A.B., M.C.-M. and R.I.-L.; Writing—original draft, M.C.-M. and R.I.-L.; Writing—review & editing, R.I.-L.; Project administration, J.A.B. All authors have read and agreed to the published version of the manuscript.

**Funding:** This research was funded by GIZ-South Africa grant number LNOB-2022. The authors would like to thank GIZ for funding the research.

**Institutional Review Board Statement:** Not applicable.

**Informed Consent Statement:** Not applicable.

**Data Availability Statement:** Data can be shared upon request.

**Conflicts of Interest:** The authors declare no conflict of interest.

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
