# Peer review of "Just Energy Transition of South Africa in a Post-COVID Era"

_sustainability, doi:10.3390/su151410854_

Round 1
Reviewer 1 Report
General comments
The authors analyze the economic effects of changing in electricity mix in South Africa, in the context of energy transition. The paper is well written in a good structure. I recommend to accept after minor reviews.
Specific comments
1. In the abstract, many all acronyms should be defined (CGE, JET, SA).
2. It is not recommended to use acronyms as keywords
3. I miss some background on CGE models. In particular, because in developing countries neoclassic approach is not the best way to represent these market. It is unclear what the option of the authors have chosen.
4. Yet, as the analysis consider energy transition, integrated models (bottom-up and top-down) are more usual. I couldn’t find any discussion on that.
5. Figure 1 and 2 are too small.
6. What is the base year of the study, referred as “t” in figure 9?
7. It is unclear if the drop in the investment, indicated in figure 9, is due to COVID-19 or energy transition.
Author Response
Reviewer 1
The authors analyze the economic effects of changing in electricity mix in South Africa, in the context of energy transition. The paper is well written in a good structure. I recommend to accept after minor reviews.
Specific comments
- In the abstract, many all acronyms should be defined (CGE, JET, SA).
Response: Done
- It is not recommended to use acronyms as keywords
Response: Done
- I miss some background on CGE models. In particular, because in developing countries neoclassic approach is not the best way to represent these market. It is unclear what the option of the authors have chosen.
Response: Unfortunately, word count limits prevent a more elaborate discussion of the CGE methodology, but the model/approach we have used is well documented in the CGE literature, including its application for developing countries. The neoclassic approach when applied in a dynamic modelling framework does not prevent sensible analysis for developing countries as, for example, temporary disequilibrium is allowed in labour markets, and various adjustment mechanisms can be calibrated using appropriate elasticity or adjustment parameters, as we believe we have done.
- Yet, as the analysis consider energy transition, integrated models (bottom-up and top-down) are more usual. I couldn’t find any discussion on that.
Response: We are not sure if this comment refers to a discussion of Integrated Assessment Models (IAMs), which are more suited to high-level macro analysis at a national level, at least as far as our understanding goes. Our analysis in this paper specifically wanted to focus on the sub-national (provincial level) dynamics of the transition in South Africa, which ruled out IAMs and dictated our choice of a regional-dynamic CGE model.
- Figure 1 and 2 are too small.
Response: Corrected. When we receive the proofs once accepted, we will work with the editorial team to improve the artwork, if needs be.
- What is the base year of the study, referred as “t” in figure 9?
Response: We added a couple of sentences in sections 4.1 and 4.2 to clarify both the base year and how the year t in the policy simulation results may be interpreted
- It is unclear if the drop in the investment, indicated in figure 9, is due to COVID-19 or energy transition.
Response: The shocks in the policy simulation run isolate and measure only the effects of the energy transition in this case – and in the case of figure 9, the change in financing costs related to coal investment projects. So, the effects of Covid-19 do not impact the policy simulations results.
Reviewer 2 Report
This is a very well written paper, with clear motivation and results. It is also a very important contribution for the African region since we dont have much academic work on the region. However, I believe the authors could add a section probably as an appendix to show the mathematics of the methodology more, because as it is, it looks as a black box. The average reader needs to know how these results appear from. I know the mathematical modelling can be extremely tedious in this area, but it would be a very importnat contribution of the paper and its citations, if you give a couple of pages length of explanation on the methodology and the software you used.
Overall this will be a very important contribution!
Author Response
Reviewer 2
This is a very well written paper, with clear motivation and results. It is also a very important contribution for the African region since we dont have much academic work on the region.
However, I believe the authors could add a section probably as an appendix to show the mathematics of the methodology more, because as it is, it looks as a black box. The average reader needs to know how these results appear from. I know the mathematical modelling can be extremely tedious in this area, but it would be a very important contribution of the paper and its citations, if you give a couple of pages length of explanation on the methodology and the software you used.
Response: Unfortunately, large-scale CGE models such as this application of UPGEM contain thousands of lines of code representing many equations describing behaviour across all components of the economy. It is typical for CGE papers, especially ones that face strict word count limits to just provide a reference to the full documentation of the model and only give a brief overview of the model’s theoretical specification in text. To do so, we added two references by Dixon et al. and Horridge et al.. We trust this will satisfy any credibility concerns.
Overall this will be a very important contribution!

Reviewer 3 Report
The present article entitled” Just greening the South African economy in a post-Covid era” is based on a good theme. As the title say just greening south African economy, but in article, no any recommendation, models are present which justify the title. In the article authors have just putted literature without any relevance.
The sentence are very lengthy and need extensive revision for example
Reframe the sentence “40-43 The effects of the COVID-19 pandemic have brought worldwide debate on how green the 40 economic recovery can and should be and whether the pandemic has accelerated the existing energy transition whilst ensuring a just transition for vulnerable groups such as 42 unskilled workers and women,
Line 69-72” the input seen as the investment in technological changes, the indirect impacts are demonstrated as losses or gains in economic welfare or job opportunities”- sentence unclear
Line-84- why repetition of “United Nations Development Programme”
Text of FIG.1-4 are not visible
-what is the recommendation or future perspective of the article
The present article entitled” Just greening the South African economy in a post-Covid era” is based on a good theme. As the title say just greening south African economy, but in article, no any recommendation, models are present which justify the title. In the article authors have just putted literature without any relevance.
The sentence are very lengthy and need extensive revision for example
Reframe the sentence “40-43 The effects of the COVID-19 pandemic have brought worldwide debate on how green the 40 economic recovery can and should be and whether the pandemic has accelerated the existing energy transition whilst ensuring a just transition for vulnerable groups such as 42 unskilled workers and women,
Line 69-72” the input seen as the investment in technological changes, the indirect impacts are demonstrated as losses or gains in economic welfare or job opportunities”- sentence unclear
Line-84- why repetition of “United Nations Development Programme”
Text of FIG.1-4 are not visible
-what is the recommendation or future perspective of the article
Author Response
Reviewer 3
The present article entitled” Just greening the South African economy in a post-Covid era” is based on a good theme. As the title say just greening south African economy, but in article, no any recommendation, models are present which justify the title. In the article authors have just putted literature without any relevance.
Response: The reviewer makes a point in the potential misuse of the concepts of “just greening” and “just energy transition” interchangeably. The literature section defines and discusses the concept of the just energy transition. We suggest changing the title to “Just Energy Transition of South Africa in a post-Covid era” to reflect the positioning and analysis of the manuscript.
The sentences are very lengthy and need extensive revision for example
Response: Thank you We have reworked the manuscript and simplified the sentences that might have the risk of being misinterpreted.
Reframe the sentence “40-43 The effects of the COVID-19 pandemic have brought worldwide debate on how green the 40 economic recovery can and should be and whether the pandemic has accelerated the existing energy transition whilst ensuring a just transition for vulnerable groups such as 42 unskilled workers and women,
Response: We have rewritten the sentence as per the suggestion. Now it reads as “The COVID-19 epidemic has sparked a major global debate about the level of environmental sustainability that can be achieved throughout the economic recovery phase. This discussion also explores how much the epidemic has accelerated the transition to sustainable energy sources while simultaneously assuring an equitable transition for vulnerable populations, particularly women and unskilled laborers.”
Line 69-72” the input seen as the investment in technological changes, the indirect impacts are demonstrated as losses or gains in economic welfare or job opportunities”- sentence unclear
Response: We have rewritten the sentence as per the suggestion. Now it reads as “The energy transition process aims at reducing greenhouse gas emissions as the main target using as input the investment in technological changes. As such, the indirect impacts are demonstrated as losses or gains in economic welfare or job opportunities.
Line-84- why repetition of “United Nations Development Programme”
Response: The second UNDP was supposed to be a reference, but has now been removed.
Text of FIG.1-4 are not visible
Response: Corrected. When we receive the proofs once accepted, we will work with the editorial team to improve the artwork, if needs be.
-what is the recommendation or future perspective of the article
Response: We have added the following paragraph in the conclusion.
The simulation results include policy suggestions for successfully managing South Africa's energy transition. These suggestions stress the importance of assisting workers and communities in the severely exposed Mpumalanga region by investing in infrastructure, retraining, and education initiatives. To address the issues that the local coal sector will face in the future, it is also suggested that the economy diversify away from coal. This can be done by making investments in renewable energy technology, promoting energy efficiency, and supporting sustainable enterprises. By setting long-term goals, clear laws, and supportive policies, governments may create a stable climate that promotes investments in renewable energy and energy efficiency. Last but not least, macroeconomic stability measures should be used to adopt efforts to reduce harmful spillover effects, like deteriorating risk and financing circumstances. With everyone benefiting from the environmental, health, economic, and financial advantages of switching to renewable energy sources, these proposals are meant to serve as a roadmap for policymakers as they work to achieve a just and sustainable energy transition in South Africa.

Round 2
Reviewer 3 Report
The article can be accepted in the presented form
The article can be accepted in the presented form